

# Information Scrambling with Conservation Laws

**Jonah Kudler-Flam[1*], Ramanjit Sohal[2,3 †] and Laimei Nie[3◇]**

**1** Kadanoff Center for Theoretical Physics, University of Chicago, IL 60637, USA
**2** Department of Physics, Princeton University, Princeton, New Jersey, 08540, USA
**3** Department of Physics and Institute for Condensed Matter Theory,
University of Illinois at Urbana-Champaign, Urbana, Illinois 61801, USA

★ jkudlerflam@uchicago.edu, † rsohal@princeton.edu, ◇ nlm@illinois.edu

## Abstract

The delocalization or *scrambling* of quantum information has emerged as a central ingredient in the understanding of thermalization in isolated quantum many-body systems. Recently, significant progress has been made analytically by modeling non-integrable systems as periodically driven systems, lacking a Hamiltonian picture, while honest Hamiltonian dynamics are frequently limited to small system sizes due to computational constraints. In this paper, we address this by investigating the role of conservation laws (including energy conservation) in the thermalization process from an information-theoretic perspective. For general non-integrable models, we use the *equilibrium approximation* to show that the maximal amount of information is scrambled (as measured by the tripartite mutual information of the time-evolution operator) at late times even when a system conserves energy. In contrast, we explicate how when a system has additional symmetries that lead to degeneracies in the spectrum, the amount of information scrambled must decrease. This general theory is exemplified in case studies of holographic conformal field theories (CFTs) and the Sachdev-Ye-Kitaev (SYK) model. Due to the large Virasoro symmetry in 1+1D CFTs, we argue that, in a sense, these holographic theories are not maximally chaotic, which is explicitly seen by the non-saturation of the second Rényi tripartite mutual information. The roles of particle-hole and $U(1)$ symmetries in the SYK model are milder due to the degeneracies being only two-fold, which we confirm explicitly at both large- and small-$N$. We reinterpret the operator entanglement in terms of the growth of local operators, connecting our results with the information scrambling described by out-of-time-ordered correlators, identifying the mechanism for suppressed scrambling from the Heisenberg perspective.



# 1 Introduction

At first glance, the emergence of thermal physics in isolated quantum systems seems to present a paradox. Consider a quantum mechanical system that has a generic Hamiltonian $H$ and is initialized in an arbitrary, finite-energy state $|\Psi\rangle$. It is generally believed that, at late times, $|\Psi\rangle$ will evolve to a state well-described[1] by a thermal state e.g. the Gibbs state $\rho_\beta := e^{-\beta H}/\text{Tr}\left[e^{-\beta H}\right]$, where $\beta$ is an *effective* temperature. It is, of course, impossible for a pure state ($|\Psi\rangle$) to actually evolve into a mixed state ($\rho_\beta$) under unitary time evolution. The solution to this apparent paradox is that the late-time state is not $\rho_\beta$ but looks like $\rho_\beta$ for sufficiently small subsystems (call this subsystem $A$). That is, if we do not have access to the information of another portion of the total system (call this subsystem $C$), the reduced state on $A$ will be very close to the reduced state of $\rho_\beta$ on $A$; all expectation values of simple operators will give their thermal value. This is the topic of the eigenstate thermalization hypothesis [1–3] and its variants [4–7].

What is the mechanism for this thermalization process? In recent years, it has become clear that quantum chaos and information scrambling play key roles in quantum thermalization in analogy with the ergodic hypothesis in the thermalization of classical systems. In the above scenario, we see that information has been largely delocalized. In particular, the information about the specific pure state $|\Psi\rangle$ is spread globally and one needs access to (close to) the entire system to determine that we are indeed dealing with a pure state that has unique microstructure distinguishing it from other pure states with equal energy. That this information is not locally present in $A$ is another way of saying that $A$ is highly entangled with $C$. We refer to

---

[1] By "well-described," we mean that expectation values of simple observables are approximately equal to their thermal value.

this phenomenon of delocalization of information due to long-range entanglement as *quantum information scrambling*.

Significant progress has been made in understanding thermalization and chaos in quantum many-body systems through the use of toy models that replace deterministic Hamiltonian evolution with random unitary evolution. Drawing from the mature mathematical field of random matrix theory, a great deal of analytic results can be derived and subsequently argued to hold for their deterministic counterparts [8]. This is one of the few analytical tools one has access to in quantum many-body chaos. However, it begs the question of if we are missing, from these toy models, key features of real Hamiltonian systems, one of which being the conservation of energy. The goal of this paper is to explore the implications of conservation laws, such as energy conservation, when considering chaos and information scrambling in quantum many-body systems. The role of conservation laws in quantum chaotic systems has been of the subject of much interest in recent years, as studied from both the perspectives of entanglement spreading, operator growth, and spectral statistics [9–17]. Our work in particular was motivated, in part, by an observation in Ref. [18] that holographic conformal field theories in $1 + 1$D do not scramble as much information as random unitary circuits, a notion that we now make precise. Before using this example as motivation, we need to introduce our method for diagnosing quantum information scrambling. A particularly straightforward and state-independent way to characterize the amount of information that is scrambled by a given Hamiltonian is by studying the tripartite mutual information in the Choi-state of the unitary evolution operator

$$|U(t)\rangle := \frac{1}{\sqrt{d}} \sum_a e^{-iE_a t} |a\rangle_1 \otimes |a^*\rangle_2, \tag{1}$$

where $d$ is the dimension of the Hilbert space, $E_a$ is the eigenenergy, and the sum is over all energy eigenvectors. This is a standard representation of a linear operator acting on Hilbert space $\mathcal{H}$ as a vector in the doubled Hilbert space $\mathcal{H}_1 \otimes \mathcal{H}_2^*$ [19,20]. By turning the unitary quantum channel into a state vector, we can diagnose the correlations between the input ($\mathcal{H}_1$) and output ($\mathcal{H}_2^*$) Hilbert spaces using entanglement measures, revealing dynamical properties of $U(t)$. The mutual information characterizes the total correlations between two density matrices

$$I^{(n)}(A,C) := S^{(n)}(\rho_A) + S^{(n)}(\rho_C) - S^{(n)}(\rho_{A\cup C}), \quad S^{(n)}(\rho) := \frac{1}{1-n} \log[\text{Tr}[\rho^n]], \tag{2}$$

where $\rho_A$ ($\rho_C$) is the reduced state resulting from partial tracing over the spatial complement of $A$ ($C$), $\rho_{A\cup C}$ is the reduced state on $A \cup C$, and $n$ is the Rényi index. As an example, we can take $A$ to be a subset of the input Hilbert space and $C$ to be a subset of the output Hilbert space. The mutual information then tells us how much information about $A$ is transmitted to $C$ under the unitary dynamics. There are other options for how the original information in $A$ is transmitted. For example, the information can be transmitted to the complement of $C$, $D$, such that the mutual information between $A$ and $D$ is large and the mutual information between $A$ and $C$ is small. Alternatively, the information may be delocalized in the sense that neither $C$ nor $D$ has complete information about $A$, only the union $C \cup D$ contains all information about $A$. In such a case, $I^{(n)}(A,C)$ and $I^{(n)}(A,D)$ are small while $I^{(n)}(A,C\cup D)$ is large. This motivates us to study the *tripartite operator mutual information* (TOMI) as a diagnostic of information scrambling of the unitary dynamics [21]

$$I_3^{(n)}(A;C,D) := I^{(n)}(A,C) + I^{(n)}(A,D) - I^{(n)}(A,C \cup D). \tag{3}$$

Due to the above argument, the TOMI will be negative only when information has been scrambled, and how negative the TOMI is characterizes how much of the information in $A$ has been

delocalized.[2] It is then natural to ask if there is a bound on the TOMI as there exists for other measures of scrambling [24,25]. First, we note that by weak subadditivity of quantum Rényi entropies [26], the Rényi mutual information in the Choi-state (1) is positive semi-definite. In a system that maximally scrambles information, one expects the mutual informations between $A$ and $C$, and between $A$ and $D$, are both zero in the long-time limit. Meanwhile, the mutual information between $A$ and $C \cup D$ (which is the entire output) is a time-independent quantity equal to twice the number of degrees of freedom in subsystem $A$. This leads us to the following fundamental lower bound

$$I_3^{(n)}(A; C, D) \geq -2 \log d_A, \tag{4}$$

where $d_A$ is the Hilbert space dimension of $A$. For the following discussion, it will be important to generalize this bound to theories with infinite-dimensional Hilbert spaces such as quantum field theories.[3] In these theories, we need to introduce a regulator, $\epsilon_{UV}$, for the Choi-state to cut off the high-energy modes

$$|U(t)\rangle_{\epsilon_{UV}} = \frac{1}{\mathscr{Z}(\epsilon_{UV})} \sum_n e^{-(it+\epsilon_{UV}/2)E_n} |n\rangle_1 \otimes |n^*\rangle_2, \tag{5}$$

where $\mathscr{Z}$ is the partition function serving as the normalization constant. With a regulator, the number of available degrees of freedom is suppressed and the lower bound is modified to[4]

$$I_3^{(n)}(A; C, D) \geq -2s_{eq}^{(n)}(\epsilon_{UV})\text{Vol}(A), \tag{6}$$

where $s_{eq}^{(n)}$ is the thermodynamic Rényi entropy density, which, at infinite temperature ($\epsilon_{UV} \to 0$) reduces back to (4).

Now, consider a generic $1+1$D conformal field theory (CFT) that is holographically dual to a theory of gravity in $2+1$D Anti-deSitter (AdS) space. Each element of this class of theories is "maximally chaotic" as determined by the out-of-time-ordered correlator (OTOC) [25,27,28] and conjectured to scramble quantum information at the fastest possible rate as measured by the approach of a subsystem to a thermal state after being perturbed [24,29,30]. One is then inclined to believe that holographic theories should saturate any bound one throws at them because they are the ultimate scramblers. It was therefore surprising to find that $1+1$D holographic CFTs do *not* saturate (6) [18]. Rather, the second Rényi TOMI saturates to a finite fraction of the lower bound. There are two plausible explanations:

1. Holographic CFTs scramble as much information as is consistent with energy conservation, i.e. no Hamiltonian system can saturate (6). This is logically possible because, to the best of our knowledge, all previously known systems that saturate (6) are periodically driven [31–33].

2. There are undriven (Hamiltonian) quantum systems that scramble information better than holographic CFTs, a phenomenon not seen in chaos diagnostics like the OTOC.

---

[2]We note that this is a different notion of information scrambling than has been studied in the context of integrable systems in Ref. [22] where the mutual information of disjoint intervals following a quench was used as a diagnostic, as in Ref. [23]. There, the nonlinear dispersion of quasiparticles led to information initially localized to broaden. However, in integrable systems, the quasiparticles are still either purely left-moving or right-moving such that the information in a non-compact system is never delocalized across the entire system, resulting in the TOMI equaling zero.

[3]Note that the TOMI is a UV finite quantity in quantum field theory that may be rigorously defined using relative entropies.

[4]This holds generically at $n = 1$. However, for $n \neq 1$ this technically has only been proven to hold in the $\epsilon_{UV} \to 0$ limit where weak subadditivity of Rényi entropy applies.

Both explanations call for scrutiny in our current methodology and understanding of quantum chaos and information scrambling. The first option, if true, implies that modeling chaotic Hamiltonian systems with random quantum systems can be very misleading. The second option counters the common belief that holographic CFTs are maximally scrambling. One of the major goals of this paper is to determine which explanation is correct. We definitively find Case 2 to be the correct explanation by explicitly computing TOMI in an energy conserving system and finding it saturates the bound. This is not to say that there is no merit to Case 1 – in fact, we find that while energy conservation does not play a role in the suppression of scrambling, other conservation laws are important. The reason why $1+1$D holographic CFTs do not saturate (6) is that, even though they have very complicated spectra, they universally contain the infinite-dimensional Virasoro symmetry. As we explain in the following sections, the energy degeneracies created by this Virasoro symmetry suppress the amount of quantum information that can be scrambled.

This paper is organized as follows. In Section 2, we review the recently introduced *equilibrated pure state* formalism. This formalism enables us to make general statements about the long-time value of TOMI in non-integrable, energy conserving systems. In particular, we argue for a mechanism of the suppression of information scrambling arising from degeneracies in the energy spectrum. In Section 3, we use the representative scrambling systems of holographic CFTs and the Sachdev-Ye-Kitaev (SYK) model to demonstrate the general conclusions of Section 2. Here, we find that adding symmetries, such as a $U(1)$ current, suppresses the scrambling in holographic CFTs even further. In contrast, we find the SYK model to saturate (6) in the large-$N$ limit. At small $N$, we use exact diagonalization to precisely confirm our general results for the TOMI of energy and charge conserving SYK models at late times. We show that approximating the late-time value of TOMI using the equilibrated pure state formalism leads to quantitatively better results than modeling the system using Haar random matrices. In Section 4, we reinterpret the TOMI in terms of the growth of local operators. We conclude that the non-saturation of TOMI directly corresponds to certain local operators remaining localized for all times. This connects our work to previous discussions on operator scrambling and chaos. Many of the technical details have been relegated to the appendices.

## 2 Equilibrated Pure States

A general approximation scheme for studying excited quantum pure states after reaching local equilibrium was put forth in Ref. [34]. We briefly review the general formalism before applying this scheme to the pure states that correspond to time evolution operators at late times. This review is, by necessity, too compressed so we encourage the interested reader to see the original work.

One of the central tools in computing von Neumann entropy in quantum systems is the replica trick [35]. We first compute the integer moments of the reduced density matrix, $\text{Tr}[\rho^n]$, then analytically continue $n$ to one to find the von Neumann entropy

$$S^{(n)}(\rho) := \frac{1}{1-n} \log[\text{Tr}[\rho^n]], \quad S_{vN}(\rho) = \lim_{n \to 1} S^{(n)}(\rho). \tag{7}$$

Even without sending $n$ to one, the Rényi entropies $S^{(n)}$ contain important information. While they do not share all of the nice properties of von Neumann entropy, they are more sensitive to certain phenomena, including conserved quantities [12–16]. We will use these sensitivities to our advantage.

In quantum mechanics and quantum field theory, it is convenient to represent the density matrix as a path integral. The path integral defines a map from the Hilbert space to itself and

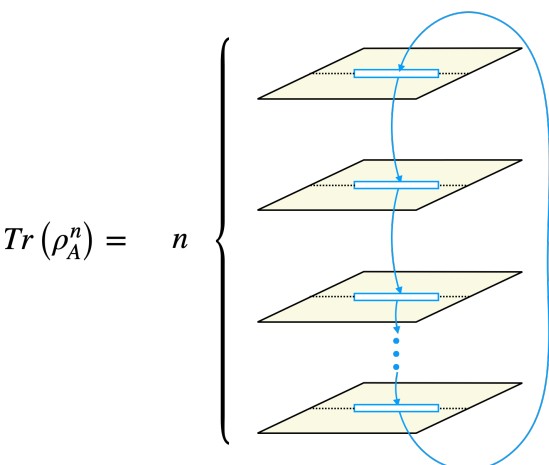

Figure 1: To compute the moments of the reduced density matrix, we prepare $n$ copies of the state using a Lorentzian path integral. The copies are glued cyclically along region $A$ (represented by the blue rectangles), implementing the matrix multiplication, but sewn to themselves in region $\bar{A}$, implementing the partial trace.

the boundary conditions determine the matrix elements of the density matrix. We define the moments of the reduced density matrix on a subregion $A$ as

$$\mathcal{Z}_n^{(A)} := \text{Tr}\left[\rho_A^n\right] = \text{Tr}_A\left[\left(\text{Tr}_{\bar{A}} U \rho_0 U^\dagger\right)^n\right], \tag{8}$$

where $\bar{A}$ is the spatial complement of $A$, $\rho_0$ is the initial state, and $U$ is the time evolution operator. This is represented as an $n$-sheeted path integral in Lorentzian time with specific boundary conditions connecting the sheets to correctly implement the trace structure (see Fig. 1). Schematically,

$$\mathcal{Z}_n^{(A)} = \int \prod_{i=1}^n D\phi_i D\phi_i' \delta(\phi_{iA}' - \phi_{(i+1)A}) \delta(\phi_{i\bar{A}}' - \phi_{i\bar{A}}) \rho_0[\phi_i, \phi_i'] e^{i\sum_{i=1}^n (I[\phi_i] - I[\phi_i'])}, \tag{9}$$

where the delta functions are imposed at time $t$, $I$ is the action, and $\phi_i$ and $\phi_i'$ are field configurations on the upper and lower edges of $A$. This partition function can then be thought of as a transition amplitude in a replicated Hilbert space $(\mathcal{H} \otimes \mathcal{H}^*)^{\otimes n}$

$$\mathcal{Z}_n^{(A)} = \langle \mathbb{1}, \eta_A \otimes e_{\bar{A}} | \left(U \otimes U^\dagger\right)^{\otimes n} | \rho_0, e \rangle, \tag{10}$$

where $\eta$ is a cyclic permutation and $e$ is the identity permutation. The cyclic permutation on $A$ implements the first delta function in (9) while the identity permutation on $\bar{A}$ implements the second delta function in (9).

Amplitudes in the replicated Hilbert space are defined in terms of the original Hilbert space as

$$\langle \mathcal{O}_1, \sigma | \mathcal{O}_2, \tau \rangle = \text{Tr}[\mathcal{O}_1 \mathcal{O}_2]^{n_1} \dots \text{Tr}[\mathcal{O}_1 \mathcal{O}_2]^{n_k}, \tag{11}$$

where $k$ is the number of cycles in permutation $\sigma^{-1} \circ \tau$ and $n_i$ is the length of the $i^{th}$ cycle. The key insight of Ref. [34] (and previously Ref. [36]) is that at late times, a special subset of field configurations in the path integral can dominate while the rest will be rapidly oscillating. Namely, these are the ones such that $\phi_i = \phi_{\sigma(i)}'$, where $\sigma$ is an element of the permutation

group $\mathcal{S}_n$. For these field configurations, the exponent in (9) vanishes so it is manifestly time-independent, implying these roughly represent an equilibrium phase.[5] Arguing that these states dominate leads to what is referred to as the *equilibrium approximation*

$$\mathcal{Z}_n^{(A)} = \frac{1}{Z_2^n} \sum_{\sigma,\tau \in \mathcal{S}_n} g^{\tau\sigma} \langle \mathbb{1}, \eta_A \otimes e_{\bar{A}} | \mathcal{I}_\alpha, \tau \rangle \langle \mathcal{I}_\alpha, \sigma | \rho_0, e \rangle, \tag{12}$$

where the metric is given by

$$g_{\tau\sigma} := \frac{\langle \mathcal{I}_\alpha, \tau | \mathcal{I}_\alpha, \sigma \rangle}{\sqrt{\langle \mathcal{I}_\alpha, \tau | \mathcal{I}_\alpha, \tau \rangle \langle \mathcal{I}_\alpha, \sigma | \mathcal{I}_\alpha, \sigma \rangle}}. \tag{13}$$

Here, $\mathcal{I}_\alpha$ is the effective identity operator where $\alpha$ labels the ensemble. For example, at infinite temperature, $\mathcal{I}_\alpha$ would just be the normal identity operator, but at finite temperature, the high energy modes will be suppressed with $\mathcal{I}_\alpha = e^{-\beta H}$. $Z_n := \text{Tr}\left[\mathcal{I}_\alpha^n\right]$ and $Z_1 := \text{Tr}[\mathcal{I}_\alpha]$ plays the role of the effective dimension on the accessible Hilbert space. This expression is further simplified to a single sum by recognizing that

$$g_{\sigma\tau} = \delta_{\sigma\tau} + O(Z_1^{-1}). \tag{14}$$

Finally, assuming $\rho_0$ is a pure state, self-consistency (i.e. $\text{Tr}\left[\left(U^\dagger \rho_0 U\right)^n\right] = 1$) ensures that

$$\langle \mathcal{I}_\alpha, \sigma | \rho_0, e \rangle = \frac{Z_2^n}{Z_1^n}, \tag{15}$$

leading to a final expression of

$$\mathcal{Z}_n^{(A)} = \frac{1}{Z_1^n} \sum_{\tau \in \mathcal{S}_n} \langle \mathbb{1}, \eta_A \otimes e_{\bar{A}} | \mathcal{I}_\alpha, \tau \rangle. \tag{16}$$

An important aspect of this equation is that the initial state, $\rho_0$, is absent. All information about the initial state is encoded in $\mathcal{I}_\alpha$, representing the ensemble that it macroscopically equilibrates to at late times.

There are two particularly important permutations in the sum that are the only ones that can dominate the partition function when $A$ and $\bar{A}$ are of unequal sizes. These are the identity and cyclic permutations. Assuming $Z_1 \gg 1$ and that the effective identity operator approximately factorizes as $\mathcal{I}_\alpha = \mathcal{I}_{A,\alpha} \otimes \mathcal{I}_{\bar{A},\alpha}$, we approximately have

$$\mathcal{Z}_n^{(A)} = \max\left[\frac{Z_{n,A}}{Z_{1,A}^n}, \frac{Z_{n,\bar{A}}}{Z_{1,\bar{A}}^n}\right], \tag{17}$$

where $Z_{n,*} := \text{Tr}\left[\mathcal{I}_*^n\right]$. In terms of entropies, this is

$$S_A^{(n)} = \min\left[S_{A,eq}^{(n)}, S_{\bar{A},eq}^{(n)}\right], \tag{18}$$

where $(eq)$ represents the thermodynamic entropy i.e. the Rényi entropy of $\rho^{(eq)} := \mathcal{I}_\alpha / \text{Tr}[\mathcal{I}_\alpha]$. This expression is manifestly invariant under $A \leftrightarrow \bar{A}$, a signature of entropies in global pure states.

---

[5]We emphasize that we are considering *late* times. At intermediate times, different regions in space-time will be characterized by distinct permutations relating the fields; the domain walls between these regions correspond to the "entanglement membrane" discussed in Appendix A and these contributions may play significant roles.

We now wish to apply this formalism to operator entanglement. The main task is identifying what the correct macroscopic ensemble is at late times. The initial state is the thermofield double

$$|\Psi_{TFD}\rangle = \frac{1}{\sqrt{\mathscr{Z}(\beta)}} \sum_n e^{-\beta E_n/2} |n\rangle_1 |n^*\rangle_2 \,. \tag{19}$$

To find the effective identity operator, we evolve the TFD state to late times and drop all matrix elements that are (on average) exponentially small in the entropy. These terms do not describe the macroscopic state and are highly theory dependent. We time evolve with $H = H_1 \otimes \mathbb{1}_2 + \mathbb{1}_1 \otimes H_2$,

$$\rho_{TFD}(t) = \frac{1}{\mathscr{Z}(\beta)} \sum_{n,m=0} e^{-it(E_n - E_m)} e^{-\beta(E_n + E_m)/2} |n\rangle\langle m|_1 \otimes |n^*\rangle\langle m^*|_2 \,. \tag{20}$$

For a sufficiently chaotic spectrum, all phases will rapidly oscillate at late times, averaging to zero, unless $E_n = E_m$. Assuming no degeneracies in the spectrum,

$$\rho_{TFD}(t \to \infty) \sim \frac{1}{\mathscr{Z}(\beta)} \sum_n e^{-\beta E_n} |n\rangle\langle n|_1 \otimes |n^*\rangle\langle n^*|_2 \,. \tag{21}$$

We stress that this state is not close to the actual state by any distance measure. Rather, it merely retains the macroscopic properties of the TFD state. This method of dephasing will further be useful when we constrain $H$ with symmetries.

This state is quite interesting as it retains memory of the correlations between the two copies without any entanglement. As it turns out, such a state was recently studied in the context of wormholes in holography [37] and dubbed the *thermomixed double* (TMD) state with effective identity operator

$$\mathcal{I}_{TMD,\beta} = \sum_n e^{-\beta E_n} |n\rangle\langle n|_1 \otimes |n^*\rangle\langle n^*|_2 \,. \tag{22}$$

In Ref. [37], it was argued that this state is the typical mixed state of the two-sided black hole and looks the same as the TFD state unless the observer has global information. It manifestly has no entanglement between the Hilbert spaces because it is written in a separable form. Moreover, the result of partial tracing over either one of the Hilbert spaces is a normal Gibbs state.

We now check the self-consistency conditions (15). First, we compute the partition functions

$$Z_k = \text{Tr}\left[\mathcal{I}_{TMD}^k\right] = \sum_n e^{-k\beta E_n} = \mathscr{Z}(k\beta) \Rightarrow \frac{Z_2^n}{Z_1^n} = \frac{\mathscr{Z}(2\beta)^n}{\mathscr{Z}(\beta)^n} \,. \tag{23}$$

To compute the LHS of (15), we note that

$$\text{Tr}\left[\rho_0 \mathcal{I}_{TMD}\right] = \text{Tr}\left[\left(\frac{1}{\mathscr{Z}(\beta)} \sum_{n,m} e^{\beta(E_n + E_m)/2} |n\rangle\langle m|_1 \otimes |n^*\rangle\langle m^*|_2\right)\left(\sum_l e^{-\beta E_l} |l\rangle\langle l|_1 \otimes |l^*\rangle\langle l^*|_2\right)\right]$$

$$= \frac{\mathscr{Z}(2\beta)}{\mathscr{Z}(\beta)} \,. \tag{24}$$

Because $\rho_0 = \rho_{TFD}$ is a pure state, $(\text{Tr}\left[\rho_0 \mathcal{I}_{TMD}\right])^k = \text{Tr}\left[(\rho_0 \mathcal{I}_{TMD})^k\right]$, so self-consistency is confirmed.

With $\mathcal{I}_{TMD}$ in hand, we can now compute operator entanglement at late times. As above, we impose that only two possible permutations can dominate the sum, $e$ and $\eta$. For $\tau = e$,

$\langle \eta_A \otimes e_{\bar{A}} | \mathcal{I}_\beta, \tau \rangle$ is given by the unnormalized Rényi purity of region $A$ while for $\tau = \eta$, it is given by the unnormalized Rényi purity of the complement region $\bar{A}$, leading to

$$S_A^{(n)}(\rho) = \min\left[ S_A^{(n)}(\rho_{TMD}), S_{\bar{A}}^{(n)}(\rho_{TMD}) \right]. \tag{25}$$

We therefore need to evaluate entropies of subsystems in the TMD state.

## 2.1 Entropies in the TMD state

First, we note that because the TMD state reduces to Gibbs state under partial trace of one of the copies, all entropies of subregions on a single side are thermal i.e. $S^{(n)} = s_{eq}^{(n)} l$ where $l$ is the size of the region. We now argue that this thermal behavior can hold even when the region has support on both copies.

Consider a region composed of an interval of size $L_A$ on one Hilbert space and $L_C$ on the other. If $H$ is translationally invariant ($[H, P_1] = [H, P_2] = 0$, $P_{1,2}$ are momentum operators on the two Hilbert spaces), then the associated TMD state is invariant under independent spatial translations on either side

$$e^{i(P_1 x_1 + P_2 x_2)} \rho_{TMD} e^{-i(P_1 x_1 + P_2 x_2)} = \rho_{TMD}. \tag{26}$$

Note that this identity does not hold for the TFD state. Because of this identity, the entropy of $A \cup C$ is independent of the relative positions of $A$ and $C$. If $L_A + L_C < L$, where $L$ is the length of one of the copies, we can always perform a spatial translation to make $A$ and $C$ disjoint.

Now, consider the entropy of disjoint intervals in the TFD state. The TFD state has spatial correlations only on length scales of over $\beta$, which we always take to be parametrically smaller than all other length scales. Therefore, the entropy of disjoint intervals, $A$ and $C$, in the TFD state is given by their thermal entropy. This is easily seen in the infinite temperature limit where the TFD state is simply a tensor product of Bell pairs. Clearly, if the regions are disjoint, the Bell pairs are not purified so the entropy is maximal (thermal). The decoherence process that takes the TFD to the TMD is an irreversible process of information loss. Then, the entropy in subsystems of the TMD should be approximately bounded below by the entropy in the TFD[6]. Because we already argued the TFD state has maximal entropy for $L_A + L_C < L$, we conclude that this must also be the case for the TMD. We provide numerical evidence for this statement in Fig. 2.

## 2.2 Conserved charges

Now, consider the case where the Hamiltonian has additional symmetry. The associated conserved charges, $Q_\alpha$, will frequently lead to degeneracies in the spectrum which is now labeled by both the energy and charge, $|n, q\rangle$ where $n$ is the energy label and $q$ is the charge label. The TFD state is rewritten as

$$\rho_{TFD}(t=0) = \frac{1}{\mathcal{Z}(\beta)} \sum_{n,m,q,r} e^{-\beta(E_n + E_m)/2} |n, q\rangle \langle m, r|_1 \otimes |n, q^*\rangle \langle m, r^*|_2. \tag{27}$$

Dephasing only allows us to drop one of the sums over energy, but leaves the sums over charge, so we no longer wind up with the TMD state for the equilibrium density matrix (normalized $\mathcal{I}_\alpha$)

$$\rho_{TFD}(t \to \infty) \sim \frac{1}{\mathcal{Z}(\beta)} \sum_{n,q,r} e^{-\beta E_n} |n, q\rangle \langle n, r|_1 \otimes |n, q^*\rangle \langle n, r^*|_2. \tag{28}$$

---

[6]We cannot make this argument rigorous because the decoherence operation and partial trace over $B \cup D$ (spatial complement of $A \cup C$) do not commute. However, we will shortly demonstrate its validity in a toy model. We expect the heuristic argument to hold generically in the thermodynamic limit.

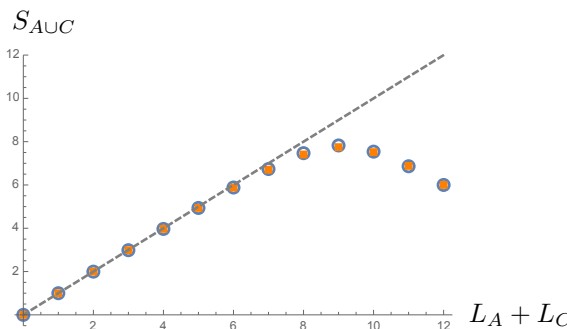

Figure 2: The von Neumman entropy (using $\log_2$) for a subsystem $A \cup C$ in the TMD state for a random subset of $(L_A + L_C)$ spins for a chaotic spin chain (blue circles) and integrable spin chain (orange squares). The dashed line is the thermal entropy. The total number of spins on each side in both models is $L = 6$, and we take $\beta = 0$ for simplicity. Each data point is obtained from averaging over all possible configurations of $A$ and $C$. The size of markers does not represent error bars; there is very small variance between individual realizations. The entropy is clearly extensive regardless of partitioning for $L_A + L_C \leq L$. Moreover, it is not sensitive to the integrability of the Hamiltonian.

This can be thought of as a charged TMD state, noting that its trace distance to the uncharged TMD state is actually large and grows with increasing degeneracy. The corresponding effective identity operator is

$$\mathcal{I}_{TMD,charged} = \sum_{n,q,r} e^{-\beta E_n} |n,q\rangle \langle n,r|_1 \otimes |n,q^*\rangle \langle n,r^*|_2. \tag{29}$$

We can check the consistency condition for this being a valid effective identity operator. First, we compute the partition function

$$Z_k = \text{Tr}\left[\mathcal{I}_{TMD,charged}^k\right] = \text{Tr}\left[\sum_{n,q,r} N_n^{k-1} e^{-k\beta E_n} |n,q\rangle \langle n,r|_1 \otimes |n,q^*\rangle \langle n,r^*|_2\right]$$

$$= \sum_n N_n^k e^{-k\beta E_n}, \tag{30}$$

where $N_n$ is the degeneracy at energy $E_n$. Next, we compute the RHS of (15)

$$\text{Tr}\left(\rho_0 \mathcal{I}_{TMD,charged}\right) = \frac{1}{\mathscr{Z}(\beta)} \text{Tr}\Bigg[\sum_{n,m,q,r,n',q',r'} e^{-\beta(E_n+E_m+2E_{n'})/2} |n,q\rangle \langle m,r|n',q'\rangle \langle n',r'|_1$$

$$\otimes |n,q^*\rangle \langle m,r^*|n',q'^*\rangle \langle n',r'^*|_2\Bigg]$$

$$= \frac{1}{\mathscr{Z}(\beta)} \sum_n N_n^2 e^{-2\beta E_n}. \tag{31}$$

$\mathcal{I}_{TMD,charged}$ then clearly passes the self-consistency check.

Due to the degeneracies in the spectrum, the equilibrium density matrix maintains a certain level of quantum coherence.[7] That is, there are non-zero off-diagonal entries in the density matrix in the energy eigenbasis. This is different than the equilibrium density matrix for

---

[7]There may be additional coherence in the density matrix due to degenerate energy level spacings, though we will not address this in the present work.

Hamiltonians without degeneracies and we claim this is the reason why the bound in (6) is not saturated. We provide a heuristic argument below and make this more mathematically precise in Appendix C.

The mutual information between $A$ and the entire output, $C \cup D$, is still $2s_{eq}^{(n)}L_A$ because this is time-independent and does not depend on the specific Hamiltonian. The difference comes from the mutual information between $A$ and subregions $C$ and $D$. Unlike the TMD state, the entropy is partition-dependent because the charged TMD state is not invariant under spatial translations of either side independently. This is due to the off-diagonal terms in the energy eigenbasis. In particular, the entropy of $A \cup C$ may be sub-thermal even when less than half the system size. As such, the late-time bipartite operator mutual information is positive and the lower bound of TOMI will no longer be saturated. This is a mechanism for suppression of scrambling of quantum information.

Let's demonstrate this mechanism with the very simple toy model of two qubits with Hamiltonian made of Pauli operators[8]

$$H = X_1 X_2 + Z_1 Z_2. \tag{32}$$

The spectrum has a degeneracy $E = \{-2, 0, 0, 2\}$. The infinite temperature TMD state is

$$\rho_{TMD} = \frac{1}{4} \sum_{i=1}^{4} |i\rangle \langle i|_1 \otimes |i\rangle \langle i|_2, \tag{33}$$

where the eigenvectors are

$$|1\rangle = \frac{1}{\sqrt{2}} (|01\rangle - |10\rangle), \quad |2\rangle = \frac{1}{\sqrt{2}} (|01\rangle + |10\rangle),$$
$$|3\rangle = \frac{1}{\sqrt{2}} (|00\rangle - |11\rangle), \quad |4\rangle = \frac{1}{\sqrt{2}} (|00\rangle + |11\rangle). \tag{34}$$

The reduced density matrix on *any* subset of two qubits, $\tilde{\rho}$, is maximally mixed

$$\tilde{\rho} = \frac{1}{4} \mathbb{1}_4, \tag{35}$$

so all Rényi entropies are thermal

$$S^{(n)}(\tilde{\rho}) = 2 \log 2. \tag{36}$$

Now consider the charged TMD state (28)

$$\rho^{(eq)} = \frac{1}{4} \big( |1\rangle \langle 1|_1 \otimes |1\rangle \langle 1|_2 + |2\rangle \langle 2|_1 \otimes |2\rangle \langle 2|_2 + |2\rangle \langle 3|_1 \otimes |2\rangle \langle 3|_2$$
$$+ |3\rangle \langle 3|_1 \otimes |3\rangle \langle 3|_2 + |3\rangle \langle 2|_1 \otimes |3\rangle \langle 2|_2 + |4\rangle \langle 4|_1 \otimes |4\rangle \langle 4|_2 \big). \tag{37}$$

The off-diagonal terms come from the energy degeneracy. The reduced density matrix on either side is still maximally mixed as expected, but now reduced density matrices of two qubits on opposite sides retain quantum coherence

$$\rho_s = \rho_a = \frac{1}{8} \begin{pmatrix} 2 & 0 & 0 & 1 \\ 0 & 2 & -1 & 0 \\ 0 & -1 & 2 & 0 \\ 1 & 0 & 0 & 2 \end{pmatrix}, \tag{38}$$

---

[8]It is not relevant that this model is integrable because we do not consider dynamics, only the equilibrium states.

where $\rho_s$ is for the two qubits directly across from each other and $\rho_a$ is for the opposite qubits. The corresponding entropies are

$$S(\rho_s = \rho_a) = 1.8113 \log 2, \quad S_2(\rho_s = \rho_a) = 1.6781 \log 2, \tag{39}$$

both of which are less than the thermal value of $2 \log 2$. Also note that the Rényi entropy is significantly more sensitive to this effect. This sensitivity of the Rényi's in many-body systems will show up again in the following section.

## 3 Case Studies

So far, the discussion has been very general. We have not chosen a specific Hamiltonian or even specified if we are working in finite-dimensional quantum mechanics or quantum field theory. The only dynamical input thus far is that we are working with theories that have sufficiently complex energy spectra in order to decohere at late times, which is a valid assumption for generic, non-integrable systems.[9] It is instructive to now specify a Hamiltonian to see the general theory at work. We have chosen holographic conformal field theories and the Sachdev-Ye-Kitaev model as case studies because they are paradigmatic models of quantum many-body chaos and are maximally chaotic as measured by the OTOC, a diagnostic of operator growth. Furthermore, both models can be enriched to incorporate additional symmetries, allowing us to isolate the impact of symmetries on information scrambling in chaotic systems.

### 3.1 Holographic Conformal Field Theories

Our working definition of a holographic conformal field theory is one that is well-described by semi-classical Einstein gravity in one higher dimension, potentially with additional quantum fields propagating on the curved space. Necessary and sufficient conditions for a CFT to be holographic have not been fully classified, though it is known that such theories are "large-$N$" and have sparse low-lying spectra [40–42].

We work with two-dimensional CFTs because computations of Rényi entropies are reduced to correlation functions of primary operators. It is certainly an interesting open problem to generalize these results to higher dimensions. We are able to do this in the von Neumann limit ($n \to 1$) by appealing to the holographic formula for entanglement entropy and the recently developed membrane theory of entanglement dynamics [36, 43–51]. However, as we will see, the von Neumann limit is not always as interesting as the higher Rényi entropies. As to not distract from our main conclusions, we relegate this computation to Appendix A.

When performing the replica trick, we glue together $n$ copies of the base manifold that represents the path integral that prepares the state of interest as in Fig. 1. This amounts to taking $n$ copies of our original theory, $CFT^{\otimes n}$, and coupling the boundary conditions. Instead of working with this replicated theory, in 2D CFT, it is convenient to gauge the discrete global symmetry by taking an orbifold by $\mathcal{S}_n$, the group that permutes the identical replicas. In the orbifold theory, $CFT^{\otimes n}/\mathcal{S}_n$, there are special operators called as *twist fields*. When taking another field around these operators, they pick up a monodromy, implementing the $\mathcal{S}_n$ symmetry. The reason that these twist fields are important for us is that correlations functions of the cyclic, $\sigma_n$, and anti-cyclic, $\bar{\sigma}_n$, twist fields in the orbifold theory are equivalent to the path integrals on the replica manifold in the original theory. For example, if we consider a single

---

[9]We note that while integrable systems do also thermalize in some sense (with their late-time states well-described by generalized Gibbs ensembles [38, 39]), they do not fully decohere and support quasiparticle excitations as a consequence of their extensive number of locally conserved quantities.

interval $[0, L_A]$ in the vacuum state, the Rényi entropies are given by

$$e^{(1-n)S^{(n)}(\rho)} := \text{Tr}[\rho^n] = \langle \sigma_n(0)\bar{\sigma}_n(L_A)\rangle. \tag{40}$$

This two-point function is computable because the twist fields behave as conformal primary fields with left and right conformal dimensions of

$$h_n = \bar{h}_n = \frac{c}{24}\left(n - \frac{1}{n}\right), \tag{41}$$

where $c$ is the central charge. Evaluating the correlation function leads to the famous result for the entanglement entropy in the ground state of the CFT, $S_{vN} = c/3 \log(L_A/\epsilon_{UV})$, where $\epsilon_{UV}$ is the UV regulator [35].

For operator entanglement in 1+1D CFT, we need to evaluate a more sophisticated correlation function of twist fields, though the approach is morally the same as the case above. First, we note that because CFTs are infinite dimensional, we need to regulate the high-energy modes to create a normalizable state dual to the unitary operator as in (5). The thermofield double state has a simple Euclidean path integral representation as an infinite strip of width $\epsilon_{UV}/2$. Forming the density matrix by gluing two strips together, we arrive at a cylinder of circumference $\epsilon_{UV}$. Therefore, to compute operator entanglement of $A = [X_1, X_2]$ on the input Hilbert space and $C = [Y_1, Y_2]$ on the output Hilbert space, we must evaluate a four-point function of twist fields on the cylinder [52]

$$\text{Tr}[\rho_{A\cup C}^n] = \langle \sigma_n(z_1, \bar{z}_1)\bar{\sigma}_n(z_2, \bar{z}_2)\sigma_n(z_3, \bar{z}_3)\bar{\sigma}_n(z_4, \bar{z}_4)\rangle_{\epsilon_{UV}}, \tag{42}$$

where the coordinates with Euclidean time $\tau$ are at

$$z_1 = \bar{z}_1 = X_1, \quad z_2 = \bar{z}_2 = X_2, \quad z_3 = \bar{z}_3^* = Y_2 + i\tau, \quad z_4 = \bar{z}_4^* = Y_1 + i\tau. \tag{43}$$

While two-point and three-point functions are fully fixed by conformal invariance, four-point functions depend on the full operator content of the given CFT. Before evaluating the correlation function, we apply a conformal map, first taking the cylinder to the complex plane, then taking the operator insertion points to the canonical choice of $\{0, 1, x, \infty\}$. The resulting correlation function is

$$\text{Tr}[\rho_{A\cup C}^n] = \left(\frac{\pi}{\epsilon_{UV}}\right)^{8h_n} \frac{x^{2h_n}\bar{x}^{2h_n}G_n(x, \bar{x})}{\left|\sinh\frac{\pi(X_1-X_2)}{\epsilon_{UV}}\sinh\frac{\pi(Y_1-Y_2)}{\epsilon_{UV}}\right|^{4h_n}}, \tag{44}$$

where $G_n$ is the conformally invariant four-point function

$$G_n(x, \bar{x}) := \langle \sigma_n(\infty)\bar{\sigma}_n(1)\sigma_n(x, \bar{x})\bar{\sigma}_n(0)\rangle_{\mathbb{C}} \tag{45}$$

and $x, \bar{x}$ are the conformally invariant cross-ratios which, after analytic continuation to Lorentzian time ($\tau \to it$), are

$$x = \frac{\sinh\left[\frac{\pi}{\epsilon_{UV}}(X_1-X_2)\right]\sinh\left[\frac{\pi}{\epsilon_{UV}}(Y_1-Y_2)\right]}{\cosh\left[\frac{\pi}{\epsilon_{UV}}(X_1-Y_2-t)\right]\cosh\left[\frac{\pi}{\epsilon_{UV}}(X_2-Y_1-t)\right]},$$

$$\bar{x} = \frac{\sinh\left[\frac{\pi}{\epsilon_{UV}}(X_1-X_2)\right]\sinh\left[\frac{\pi}{\epsilon_{UV}}(Y_1-Y_2)\right]}{\cosh\left[\frac{\pi}{\epsilon_{UV}}(X_1-Y_2+t)\right]\cosh\left[\frac{\pi}{\epsilon_{UV}}(X_2-Y_1+t)\right]}. \tag{46}$$

Note that after analytic continuation, $x$ and $\bar{x}$ are independent real parameters. The individual entropies for $A$ and $C$ are time independent and equal to their thermal values because they

only probe a single side of the thermofield double state. The Rényi mutual information then simplifies to

$$I^{(n)}(A,C) = \frac{1}{n-1} \log\left[ x^{2h_n} \bar{x}^{2\bar{h}_n} G_n(x,\bar{x}) \right]. \tag{47}$$

In general, the function $G_n$ is difficult to evaluate as it depends on the full operator content of the theory. We will be able to do this in certain cases, both when the CFT has a twist gap (no extended symmetry algebra beyond Virasoro) and when the CFT has an additional $\mathfrak{u}_k(1)$ Kac-Moody symmetry.

**Von Neumann Limit**   We can always expand $G_n$ in a basis of Virasoro conformal blocks, $\mathcal{F}_T$, as

$$G_n(x,\bar{x}) = \sum_p |C_{nnp}|^2 \mathcal{F}_T(h_n, h_p, c, x) \bar{\mathcal{F}}_{\bar{T}}(\bar{h}_n, \bar{h}_p, \bar{c}, \bar{x}), \tag{48}$$

where the sum is over all Virasoro primary fields and $C_{nnp}$ is the operator product expansion (OPE) coefficient between two twist fields and the Virasoro primary. At large central charge, the Virasoro blocks approximately exponentiate [53]

$$\mathcal{F}_T(h_n, h_p, c, x) \simeq e^{-\frac{c}{6} f(x, h_n, h_p)}. \tag{49}$$

Therefore, it is a good approximation to only keep a single term in the sum in the $c \to \infty$ limit.

When $n \sim 1$, the twist fields become light i.e. $h_n \ll c$. It is well-known that when all operators in the correlation function are light at large $c$, the dominant Virasoro conformal block reduces to the global conformal block [54], meaning all exchanges of Virasoro descendants are subleading. This has a simple expression in terms of hypergeometric functions

$$\mathcal{F}_T(h_n, h_p, c, x) = (1-x)^{h_p - 2h_n} {}_2F_1(h_p, h_p, 2h_p, 1-x) + O\left(c^{-1}\right). \tag{50}$$

Among these global conformal blocks, the vacuum block ($h_p = 0$) will dominate because it has the lowest conformal dimension. We therefore arrive at

$$G_{n \sim 1} \simeq \min\left[ x\bar{x}, (1-x)(1-\bar{x}) \right]^{-2h_n}. \tag{51}$$

Here, we have included the $x\bar{x}$ term which is the dominant (identity) conformal block in the cross channel ($x \to 0$).

While the full time dependence of the operator entanglement is interesting, we focus on the late-time behavior. In this limit, for $I(A,C)$ ($I(A,D)$), we have $\bar{x}, (1-x) \to 0$ ($x, (1-\bar{x}) \to 0$) with $\bar{x}(x)$ going to zero faster, so the first term in the minimization dominates. From (47), this immediately leads to trivial bipartite mutual information

$$I(A,C) = I(A,D) = O\left(c^{-1}\right). \tag{52}$$

On the other hand, $I(A, C \cup D)$ is time-independent, so we have

$$I_3(t \to \infty) = -\frac{2\pi c L_A}{3\epsilon_{UV}} + O\left(c^{-1}\right), \tag{53}$$

where we have defined $L_A := X_2 - X_1$. This saturates (6).

Adding a $\mathfrak{u}_k(1)$ symmetry to the holographic CFT corresponds to adding a level $k$ Chern-Simons field to the AdS bulk action. We can ask how this will change the answer. First, we note that the addition of this chiral primary field means that other conformal blocks can become

important, namely the $\mathfrak{u}_k(1)$ descendants of the vacuum state. We can repackage the Virasoro conformal blocks into $Vir \times \mathfrak{u}_k(1)$ blocks to take into account all of these contributions. Happily, these extended blocks also take a simple form at large $c$ due to a factorization into Virasoro and $\mathfrak{u}_k(1)$, $\mathcal{F}_J$, blocks [55]

$$\mathcal{F}_{T,J}(h_n, h_p, q, k, c, z) = \mathcal{F}_T\left(h_n - \frac{q_n^2}{2k}, h_p, c-1, x\right)\mathcal{F}_J(q_n, k, x), \tag{54}$$

where the $q_n$'s are the $\mathfrak{u}_k(1)$ charges of the operators. The $\mathfrak{u}_k(1)$ block is

$$\mathcal{F}_J(k, q_n, x) = x^{-\frac{q_n^2}{k}}(1-z)^{\frac{q_n^2}{k}}. \tag{55}$$

Twist operators are uncharged under the $\mathfrak{u}_k(1)$ ($q_n = 0$), so the only effect is taking $c \to c-1$ in the Virasoro block. This gives the same answer (53).

**Rényi Entropy**  More interesting than the von Neumann limit is the Rényi entropy. The integer Rényi entropies are more sensitive than the von Neumann entropy to the finer details in the entanglement spectrum. This, for example, was demonstrated in Ref. [56] where it was noted that $I_3$ saturates the lower bound in the thermodynamic limit even when the unitary operator only scrambles in a subspace of the Hilbert space while the second Rényi entropy does not.

We will focus on the second Rényi entropy from now on. Using a conformal mapping from the two-sheeted Riemann surface to the torus [57, 58], we can express the operator mutual information in terms of the torus partition function for any 2D CFT

$$I^{(2)}(A, C)(t) = \log\left[2^{-2c/3}|x|^{c/6}|1-x|^{-c/12}\mathcal{Z}(\tau, \bar{\tau})\right]. \tag{56}$$

The moduli of the torus are related to the cross-ratios as

$$\tau = i\frac{K(1-x)}{K(x)}, \quad \bar{\tau} = -i\frac{K(1-\bar{x})}{K(\bar{x})}, \tag{57}$$

where $K(x)$ is an elliptic integral of the first kind. Knowledge of the operator entanglement has been reduced to knowledge of the torus partition function, a regularly studied object in conformal field theory. The CFT partition function is

$$\mathcal{Z}(\tau, \bar{\tau}) = \text{Tr}\left[q^{L_0 - \frac{c}{24}}\bar{q}^{\bar{L}_0 - \frac{c}{24}}\right], \quad q = e^{2\pi i \tau}, \quad \bar{q} = e^{-2\pi i \bar{\tau}}. \tag{58}$$

The late-time limit of BOMI only requires us to understand the structure of the trace when the elliptic nomes $(q, \bar{q})$ are exponentially close to zero and one. This allows us to compute entanglement measures even without complete knowledge of the operator content of the CFT. In the mixed limit $x \to 0, \bar{x} \to 1 - e^{-\frac{2\pi L_A}{\epsilon_{UV}}}$, the partition function will be dominated by all states with $h = 0$

$$\mathcal{Z}(\tau, \bar{\tau}) = q^{-\frac{c}{24}}\sum_{(h,\bar{h})=(0,\bar{h})}\bar{q}^{\bar{L}_0 - \frac{c}{24}}. \tag{59}$$

The sum is over all anti-chiral states. By definition, this sum is the vacuum character of the chiral algebra, $\mathfrak{A}$,

$$\mathcal{Z}(\tau, \bar{\tau}) = q^{-\frac{c}{24}}\bar{\chi}_{\mathfrak{A}}(\bar{q}). \tag{60}$$

When the full chiral algebra is Virasoro, this is simply the vacuum Virasoro character

$$\bar{\chi}_T(\bar{q}) = \frac{\bar{q}^{-\frac{c-1}{24}}(1-\bar{q})}{\eta(\bar{\tau})}. \tag{61}$$

When there is also a $\mathfrak{u}_k(1)$ symmetry, we have

$$\bar{\chi}_{T,J}(\bar{q}) = \frac{\bar{q}^{-\frac{c-2}{24}}(1-\bar{q})}{\eta(\bar{\tau})^2}. \tag{62}$$

Because $\bar{\tau} \sim 0$ and imaginary, it is useful to recall the modular properties of the Dedekind eta function

$$\eta\left(-\frac{1}{\bar{\tau}}\right) = \sqrt{i\bar{\tau}}\,\eta(\bar{\tau}). \tag{63}$$

$\bar{q}' = e^{2\pi i/\bar{\tau}}$ is exponentially small, so we can take just the first term in the $q$-series

$$\eta(\bar{\tau}) \simeq (i\bar{\tau})^{-1/2} e^{\pi i/12\bar{\tau}} \simeq \sqrt{-\frac{\log\left(\frac{1-\bar{x}}{16}\right)}{\pi}}\left(\frac{1-\bar{x}}{16}\right)^{1/12}. \tag{64}$$

Using the late-time value for the cross-ratio, we find at leading order

$$I_T^{(2)}(A,C) = \frac{\pi(c+2)L_A}{12\epsilon_{UV}} \tag{65}$$

for $c > 1$ theories with a twist gap and

$$I_{T,J}^{(2)}(A,C) = \frac{\pi(c+4)L_A}{12\epsilon_{UV}} \tag{66}$$

for theories with an additional $\mathfrak{u}_k(1)$. Analogous results hold for $I^{(2)}(A,D)$, while $I^{(2)}(A,C\cup D)$ is time independent and theory independent. There are various interesting features of these equations. Most importantly, they are non-zero. This is surprising because it means that not all information is delocalized under unitary time evolution as one would expect for a chaotic theory (and suggested by the von Neumann limit). This is not a subleading effect in the $1/c$ expansion, creating tension with the lore that large-$c$ holographic CFTs are maximally scrambling. Next, note that the saturation value scales linearly with the size of region $A$. Thus, an extensive amount of information remains localized. Finally, the $\mathfrak{u}_k(1)$ symmetry implements a subleading effect[10], increasing the saturation value by $\frac{\pi L_A}{6\epsilon_{UV}}$.

Combining the results for $I^{(2)}$, we find the tripartite mutual information to have the following saturation values

$$I_{3,T}^{(2)}(A;C,D) = -\frac{\pi(c-1)L_A}{3\epsilon_{UV}}, \quad I_{3,T,J}^{(2)}(A;C,D) = -\frac{\pi(c-2)L_A}{3\epsilon_{UV}}. \tag{67}$$

Clearly, neither theory saturates the lower bound (6) and the additional symmetry further suppresses the scrambling.

---

[10]This can be made into a leading order effect if we add an $O(c)$ number of $\mathfrak{u}_k(1)$ currents.

**Degeneracies in Verma Modules**    To connect with our earlier discussion, we recall the structure of the Hilbert space of 2D CFTs. 2D CFTs are organized into Verma modules which are representations of the $Vir \times \overline{Vir}$ symmetry. The Verma modules are labeled by conformal primary fields which generate the highest weight states

$$|\phi\rangle = \phi(0)|0\rangle, \tag{68}$$

where $|0\rangle$ is the unique conformally invariant vacuum state that is annhilated by the Virasoro generators

$$L_n |0\rangle = \bar{L}_n |0\rangle = 0, \quad \forall n \geq -1. \tag{69}$$

The energies of the highest weight states are

$$E_\phi = h_\phi + \bar{h}_\phi - \frac{c}{12}, \tag{70}$$

where $h_\phi$ ($\bar{h}_\phi$) are the left (right) conformal weights of the primary operator. All other states (descendants) in the Hilbert space can be generated from the highest weight states by acting with the Virasoro generators

$$|\phi; k_1, \bar{k}_1, \ldots k_n, \bar{k}_n\rangle := L_{-1}^{k_1} \bar{L}_{-1}^{\bar{k}_1} \ldots L_{-n}^{k_n} \bar{L}_{-n}^{\bar{k}_n} |\phi\rangle. \tag{71}$$

Using the Virasoro algebra,

$$[L_n, L_m] = (n-m)L_{n+m} + \frac{c}{12}n(n^2-1)\delta_{n+m,0}, \tag{72}$$

one can show that the descendant states are also energy eigenstates, but with energy

$$E_{\phi,\{k\}} = h_\phi + \bar{h}_\phi + \sum_j j(k_j + \bar{k}_j) - \frac{c}{12}. \tag{73}$$

This structure is crucial because it implies universal large degeneracies in the spectrum. This degeneracy comes from the sum which is a sum of integers. For simplicity, we focus only on the holomorphic sector and define the level as $N = \sum_j jk_j$. The degeneracy at level $N$ will be equal to the number of integer partitions of $N$. This grows exponentially quickly with $N$ and becomes even larger when we account for the anti-holomorphic sector. Therefore, even chaotic (e.g. holographic) CFTs that may have chaotic-looking spectra of conformal primaries, will have massive degeneracies within each Verma module that limit the ability of the system to decohere and scramble quantum information.

Next, we consider the effect of including a Kac-Moody current $J^a$ where the index labels the generators of the affine Lie algebra. Without the symmetry, we assumed that conformal primaries of different weight were not correlated in the sense that $h_{\phi_1} - h_{\phi_2} \notin \mathbb{Z}$. This is naturally the case for a chaotic spectrum and implies that degeneracies only occur within individual Verma modules. This is not the case with Kac-Moody symmetry because the modes of the current operator will connect various Verma modules. The simplest version of this statement is that the vacuum Verma module is connected to the Verma modules corresponding to the currents, which are conformal primaries, simply by the action of the current modes

$$|J^a\rangle = J^a_{-1} |0\rangle. \tag{74}$$

The connections between Verma modules clearly enhances the degeneracies in the system, further restricting decoherence. We attribute these additional degeneracies to the observed suppression of scrambling in (67).

## 3.2 Sachdev-Ye-Kitaev Models

Our computation of the operator mutual information in holographic CFTs provides a concrete demonstration of the general picture developed in Sec. 2 of how degeneracies in the spectrum can inhibit information scrambling. It would be desirable, however, to see this effect in a specific microscopic system. To that end, we next turn our attention to the Sachdev-Ye-Kitaev (SYK) models [59, 60].

The original SYK model [60, 61] describes a zero-dimensional system of $N$ Majorana fermions coupled via random all-to-all interactions, as governed by the Hamiltonian,

$$H = \sum_{1 \leq i < j < k < l \leq N} J_{ijkl} \chi_i \chi_j \chi_k \chi_l. \tag{75}$$

Here, the $\chi_i$ are Majorana fermions satisfying

$$\{\chi_i, \chi_j\} = \delta_{ij}. \tag{76}$$

The random couplings $J_{i_1 \dots i_q}$ are fully antisymmetric in their indices and are sampled from a Gaussian distribution with mean and variance,

$$\overline{J_{i_1 \dots i_q}} = 0, \qquad \overline{J_{ijkl}^2} = \frac{3! J^2}{N^3}. \tag{77}$$

A more general Hamiltonian involving interactions with $q$ Majorana operators may be considered. We will restrict ourselves to $q = 4$ for concreteness and due to the symmetry-enforced degeneracies when $q$ is a multiple of four, as discussed below. In spite of its high degree of non-locality, the SYK model is in fact exactly solvable in the large-$N$ limit, a feature which we will exploit in the following. Indeed, at low energies, the SYK model exhibits an emergent conformal symmetry (which is both explicitly and spontaneously broken), allowing for an analytic computation of the OTOC, which can be shown to saturate the chaos bound. These features are reminiscent of holographic systems and, indeed, the SYK model has a low energy sector well-described by Jackiw-Teitelboim gravity. For these reasons, the SYK model serves as an ideal playground to contrast with our computations in 2D holographic CFTs above. As far as symmetries are concerned, the SYK Hamiltonian preserves fermion parity and possesses a particle-hole conjugation symmetry, as we will review below.

The SYK model can be readily generalized to incorporate additional global symmetries. By considering a system of complex fermions in place of Majorana fermions, we can define a model with a conserved $U(1)$ charge. The complex SYK (cSYK) model [62, 63] of $N_c$ complex fermions $c_i$ is defined by the Hamiltonian

$$H_c = \sum_{\substack{1 \leq i < j \leq N_c \\ 1 \leq k < l \leq N_c}} J_{ij;kl} c_i^\dagger c_j^\dagger c_k c_l - \mu \sum_i c_i^\dagger c_i, \tag{78}$$

where $\mu$ is a chemical potential and the random complex couplings have mean and variance,

$$\overline{J_{ij;kl}} = 0, \quad \overline{J_{ij;kl}^2} = \frac{2 J^2}{N_c^3}. \tag{79}$$

The couplings are fully anti-symmetric under exchange of any pair of $i$ indices or any pair of $j$ indices. Requiring that the Hamiltonian be Hermitian enforces

$$J_{ij;kl} = J_{kl;ij}^*. \tag{80}$$

We can define a particle-hole transformation:

$$c_i \leftrightarrow c_i^\dagger, \quad J_{ij;kl} \to J_{ij;kl}^*.$$ (81)

As written, the Hamiltonian given in Eq. (78) is not symmetric under this transformation for $\mu = 0$ due to the additional terms that arise when commuting the $c_i$ and $c_i^\dagger$ operators past one another. One can rectify this by anti-symmetrizing the SYK interaction to obtain the modified Hamiltonian [63,64],

$$\tilde{H}_c = \sum_{\substack{1 \le i < j \le N_c \\ 1 \le k < l \le N_c}} J_{ij;kl} \left( c_i^\dagger c_j^\dagger c_k c_l + \frac{1}{2} \left[ \delta_{ik} c_j^\dagger c_l - \delta_{il} c_j^\dagger c_k - \delta_{jk} c_i^\dagger c_l + \delta_{jl} c_i^\dagger c_k + \frac{1}{2} \delta_{il} \delta_{jk} - \frac{1}{2} \delta_{ij} \delta_{kl} \right] \right)$$
$$- \mu \sum_i \left( c_i^\dagger c_i - \frac{1}{2} \right).$$ (82)

The modified Hamiltonian $\tilde{H}$ is symmetric under the transformation of Eq. (81) for $\mu = 0$. However, $H$ and $\tilde{H}$ differ by terms which will be of subleading order in $N_c$ when we take the large-$N_c$ limit. Hence, both $H$ and $\tilde{H}$ are described by the same effective action in the large-$N_c$ limit, which is invariant under the particle-hole transformation for $\mu = 0$.

Our goal in the balance of this section will be to compute the Rényi TOMI of both the SYK and cSYK models. The von Neumann TOMI was previously computed in the SYK model at finite-$N$ in Ref. [21], where it was found that it saturates near the Haar random value. More recently, a calculation of the Rényi TOMI at large but finite $N$ has been carried out for the Brownian SYK model, in which the couplings are given a random time dependence, corresponding to random kicks of the system [65]. It was found that this model saturates the bound of the TOMI. We will find that this remains true in the case of time-independent couplings, implying that energy conservation *does not* play a role in limiting information scrambling. This in of itself is an interesting result: it echoes our previous observation of holographic CFTs, where the Rényi TOMI fails to saturate the lower bound due to spectral degeneracies, rather than energy conservation.

Let us now turn to our primary goal, which is to understand how degeneracies in the spectra of the SYK and cSYK models suppress information scrambling. Indeed, as we will review, both models exhibit symmetry enforced double degeneracies, depending on the value of $N$ ($N_c$). Unlike the computations of the preceding section, we are unable to derive an analytic expression for the TOMI of the SYK models for arbitrary $N$. Our strategy will therefore be to instead numerically analyze the SYK models in the large-$N$ and small-$N$ limits. The former limit is amenable to path-integral methods while the latter can be handled using exact diagonalization. We will find that the TOMI bound is saturated in the large-$N$ limit for both the SYK and cSYK models, while for small $N$ we find a clear suppression of the TOMI in the cases where the energy spectrum exhibits degeneracies. Furthermore, the values of TOMI for small-$N$ models are in quantitative agreement with the equilibrated pure state formalism reviewed in Section 2 (once we take into account additional subleading corrections).

**Symmetries and Degeneracies**   Before proceeding to our calculations, we begin by reviewing the symmetries of these SYK models and their connections to degeneracies in their spectra [64,66–69]. Starting with the Majorana SYK model, it is clear to see that the Hamiltonian of Eq. (75) possesses a $\mathbb{Z}_2$ fermion parity symmetry. Explicitly, we can define the fermion parity as

$$(-1)^F = 2^{N/2} i^{N/2} \prod_{i=1}^N \chi_i.$$ (83)

We then have that $[(-1)^F, H] = 0$, from which it follows that the eigenstates of $H$ fall into two superselection sectors, corresponding to even and odd fermion parity. We are interested in the degeneracies in the spectrum of $H$ as a consequence of this symmetry. To this end, we can define an antiunitary operator

$$P = 2^{N/2} K \prod_{i=1}^{N/2} \chi_{2i}, \tag{84}$$

where $K$ is the complex conjugation operator and we choose a basis such that

$$K \chi_{2i} K = \chi_{2i}, \quad K \chi_{2i+1} K = -\chi_{2i+1}. \tag{85}$$

It is readily checked that $P^2 = (-1)^{(N/2)((N/2)-1)/2}$, and so

$$P \chi_i P^{-1} = (-1)^{N/2-1} \chi_i. \tag{86}$$

Hence,

$$P(-1)^F P^{-1} = (-1)^{N/2} (-1)^F, \tag{87}$$

while

$$PHP^{-1} = H. \tag{88}$$

Hence $H$ and $P$ commute. Now, if $N/2$ is odd, then $P$ maps odd parity states to even parity states and vice versa. Hence, for every even parity eigenstate of $H$, there is an odd parity state with the same energy, implying that the entire spectrum is doubly degenerate. Now, if $(N/2) = 2 \mod 4$, then $[P, (-1)^F] = 0$, so that $P$ maps each parity sector onto itself. However, $P^2 = -1$, and so following the standard Kramers' degeneracy argument, an eigenstate $|\psi\rangle$ of $H$ must be orthogonal to $P|\psi\rangle$. Hence the spectrum is again doubly degenerate. Finally, if $(N/2) = 0 \mod 4$, we still have $[P, (-1)^F] = 0$, but now $P^2 = 1$. In this case, the spectrum has no protected degeneracies. Recapitulating, the Majorana SYK model has a doubly degenerate spectrum if $N \neq 0 \mod 8$ and a nondegenerate spectrum if $N = 0 \mod 8$. Note that, if we instead considered a generalized version of the SYK Hamiltonian with $q$-Majorana interactions with $q$ a multiple of four, then $H$ and $P$ would still commute and the same arguments would hold.

The same analysis can be applied to the cSYK model. In this case, we have a conserved $U(1)$ charge,

$$Q = \sum_{i=1}^{N_c} \left( c_i^\dagger c_i - \frac{1}{2} \right). \tag{89}$$

Let us restrict ourselves to the case in which the chemical potential is tuned to the particle-hole symmetric point. We can again define the antiunitary particle-hole conjugation operator which, in terms of these complex fermions, takes the form

$$P = K \prod_{i=1}^{N_c} (c_i + c_i^\dagger). \tag{90}$$

Note that we take the complex fermions to be invariant under complex conjugation: $K c_i K = c_i$, $K c_i^\dagger K = c_i^\dagger$. We have that $P^2 = (-1)^{N_c(N_c-1)/2}$ and

$$P c_i P^{-1} = (-1)^{N_c-1} c_i^\dagger \tag{91}$$

$$P c_i^\dagger P^{-1} = (-1)^{N_c-1} c_i. \tag{92}$$

Hence,

$$PQP^{-1} = -Q, \tag{93}$$

while $P$ and $\tilde{H}$ commute for $\mu = 0$. Hence, if $|\psi\rangle$ is an eigenstate of $\tilde{H}$ with charge $Q$, then $P|\psi\rangle$ is also an eigenstate with the same energy and charge $-Q$. Note that we can have $P|\psi\rangle \propto |\psi\rangle$ only if $Q = 0$. If $N_c$ is odd, then $Q$ must take half-integer values, and so $|\psi\rangle$ and $P|\psi\rangle$ are always distinct, implying that the entire spectrum of $\tilde{H}$ is doubly degenerate. On the other hand, if $N_c$ is even, while states in charge sectors with $Q \neq 0$ always have degenerate particle-hole conjugate partners in charge sectors $-Q$, there are two possibilities for the degeneracy of the $Q = 0$ sector. If $N_c \bmod 4 = 2, 3$, then $P^2 = -1$, and so a state $|\psi\rangle$ with $Q = 0$ will be orthogonal to the state $P|\psi\rangle$ by the usual Kramers' argument. On the other hand, if $N_c \bmod 4 = 0, 1$, then $P^2 = +1$, and so there is no symmetry protected degeneracy of the $Q = 0$ sector.

We now seek to investigate the impact of these degeneracies on scrambling in these systems. Our expectation is that parameters yielding a greater number of degeneracies should result in a suppression of information scrambling.

### 3.2.1 Large-$N$

We now proceed to compute the Rényi TOMI in these SYK models, focusing first on their large-$N$ limit. For reference, we repeat here the definition of the second-order Rényi TOMI:

$$I_3^{(2)}(A; C, D) = S_A^{(2)} + S_C^{(2)} + S_D^{(2)} - S_{A\cup C}^{(2)} - S_{A\cup D}^{(2)} - S_{C\cup D}^{(2)} + S_{A\cup C\cup D}^{(2)}. \tag{94}$$

Although the Rényi entropies may formally be cast as correlation functions of twist operators as in the above CFT calculation, the absence of conformal invariance in the present case means we have no analytical handle with which to compute these correlators. Instead, we are forced to express the entropies as path integrals in a replicated Hilbert space with twisted boundary conditions for the fermions. Fortunately, these quantities can be numerically evaluated *exactly* in the large-$N$ limit [70–74], as we now illustrate.

Our starting point is again the state dual to the time evolution operator:

$$|U(t)\rangle_\epsilon = \frac{1}{\sqrt{\mathscr{Z}(\epsilon)}} e^{-(it+\epsilon/2)H_1} |I\rangle_{12}, \tag{95}$$

where $\mathscr{Z}(\epsilon)$ is the partition function at inverse temperature $\epsilon$. Here we have chosen to place all the time evolution on the output Hilbert space and we have defined $|I\rangle_{12}$ to be the infinite-temperature thermofield double state.[11] We wish to write down a general expression for the second Rényi entropy of an arbitrary subregion with support both in the output and input Hilbert spaces. To that end, let us bipartition the output Hilbert space into $X_1$ and $\overline{X}_1$, as well as the input Hilbert space into $Y_2$ and $\overline{Y}_2$ [see Fig. 3(a)]. Note that the bar in $\overline{X}_1$ indicates the complement of $X_1$ within the output Hilbert space, not the full doubled Hilbert space. In the following, we will also denote by $Y_1$ the region in the output Hilbert space corresponding to the region $Y_2$ in the input Hilbert space [see Fig. 3(b)].

We are interested in computing the second Rényi entropy

$$S_{X_1 \cup Y_2}^{(2)} = -\log \text{Tr}\left[\rho_{X_1 \cup Y_2}^2\right], \tag{96}$$

---

[11]One convenient definition of $|I\rangle_{12}$ is to take it to be the state satisfying $(\chi_i^1 + i\chi_i^2)|I\rangle_{12} = 0$, where $\chi_i^a$ is a fermion in Hilbert space $a = 1, 2$ [75]. By forming complex fermions from the Majorana fermions as $c_i^1 = (\chi_{2i-1}^1 + i\chi_{2i}^1)/2$ and $c_i^2 = (\chi_{2i-1}^1 - i\chi_{2i}^1)/2$, one can see that this corresponds to a product of $N/2$ Bell pairs in the $c_{i,a}^\dagger c_{i,a}$ eigenbasis: $|I\rangle_{12} \propto \prod_i (|0\rangle_{i,1}|1\rangle_{i,2} + |1\rangle_{i,1}|0\rangle_{i,2})$.

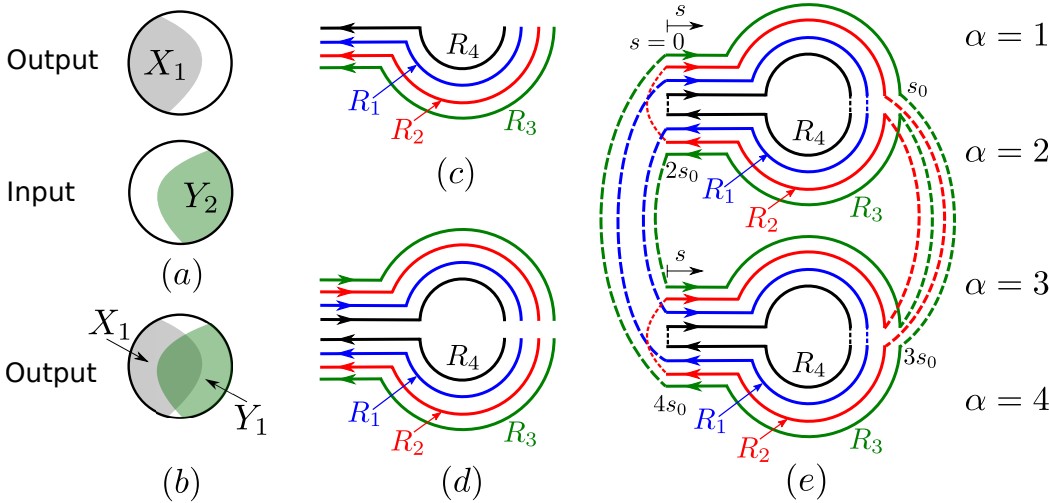

Figure 3: (a) Setup of the TFD state for the SYK model. The two circles represent the the output and input Hilbert spaces and the corresponding spatial bipartitions into $X_1/\overline{X}_1$ and $Y_1/\overline{Y}_1$, respectively. (b) The output Hilbert space, with $Y_1$, the region corresponding to $Y_2$ in the input Hilbert space, depicted alongside $X_1$. (c) A contour representation of the TFD state. Starting from the maximally entangled state $|I\rangle_{12}$, the output Hilbert space is evolved for an imaginary time $\epsilon/2$ (represented by the curved lines) and then by a real time $t$ (represented by horizontal lines). The arrows indicate the direction of real time evolution. We depict the evolution for the fermions in each of the spatial subregions $R_i$, defined in Eq. (99). (d) Contour representation of the full TFD density matrix. (e) Contour representation of the Rényi entropy of Eq. (98). The dashed lines indicate the traces for each of the $R_i$ regions. Also indicated is our parameterization of the contour in terms of $s$, where we have set $s_0 = t + \epsilon/2$. Lastly, the number $\alpha$ labels the four contour segments (see Appendix B).

where $\rho_{X_1 \cup Y_2}$ is the reduced density matrix for the region $X_1 \cup Y_2$. Explicitly,

$$\rho_{X_1 \cup Y_2} = \frac{1}{\mathscr{Z}(\epsilon)} \text{Tr}_{\overline{X_1 \cup Y_2}}\left[ e^{-(it+\frac{\epsilon}{2})H_1} |I\rangle_{12\ 12}\langle I| e^{(it+\frac{\epsilon}{2})H_1} \right]. \tag{97}$$

For convenience, we set $U_1 := e^{-(it+\epsilon/2)H_1}$. Now, since $|I\rangle_{12}$ is a product of Bell pairs, we have that $\text{Tr}_{\overline{Y}_2}\left( |I\rangle_{12\ 12}\langle I| \right) = |I\rangle_{Y_1 Y_2\ Y_1 Y_2}\langle I|$, where $|I\rangle_{Y_1 Y_2}$ is the infinite temperature TFD state for the region $Y_1$. Using this fact and by inserting complete bases of states, we can express the trace as

$$\text{Tr}_{X_1 \cup Y_2}\rho_{X_1 \cup Y_2}^2 = \mathscr{Z}(\epsilon)^{-2} \sum_{\substack{x,x',\overline{x},\overline{x}' \\ y,y',\overline{y},\overline{y}'}} \left[ \langle x\overline{x}| U_1 |y\overline{y}\rangle \langle y'\overline{y}| U_1^\dagger |x'\overline{x}\rangle \langle x'\overline{x}'| U_1 |y'\overline{y}'\rangle \langle y\overline{y}'| U_1^\dagger |x\overline{x}'\rangle \right]. \tag{98}$$

Here, $|x\rangle$ and $|y\rangle$ label complete bases of states for the subspaces $X_1$ and $Y_1$, respectively, while $|\overline{x}\rangle$ and $|\overline{y}\rangle$ label complete bases of states for $\overline{X}_1$ and $\overline{Y}_1$, respectively. This expression for the Rényi entropy is formally equivalent to those computed for holographic CFTs above. We have presented it in this cumbersome form to emphasize that this Rényi entropy can be expressed as a Schwinger-Keldysh path integral of just the output Hilbert space with appropriately twisted boundary conditions for the fermions, as presented in Fig. 3.

Indeed, to unpack this expression and Fig. 3, we first define the following four subregions

of the output Hilbert space:

$$R_1 := X_1 \backslash (X_1 \cap Y_1), \quad R_3 := X_1 \cap Y_1,$$
$$R_2 := Y_1 \backslash (X_1 \cap Y_1), \quad R_4 := \overline{X_1} \cap \overline{Y_1}. \tag{99}$$

The fermions in each of these regions will obey distinct boundary conditions. Now, the thermofield double state can be pictorially represented as in Fig. 3(c), where the semicircle and horizontal line correspond to imaginary time evolution followed by real time evolution, respectively, of just the output Hilbert space from the state $|I\rangle_{12}$. The density matrix is represented by two copies of this contour, as shown in Fig. 3(d). The square of the density matrix is then given by the contour of Fig. 3(e), consisting of two replicas. Evaluating the second Rényi entropy corresponds to appropriately tracing over the boundary conditions of the fermions in each of the regions $R_i$, represented by the dashed lines in Fig. 3(e), which can be read off from the explicit form of the trace in Eq. (98). Each of the four overlaps in Eq. (98) corresponds to one of the TFD contours in Fig. 3(e) – matching up the bras and kets of the overlaps tells us how to connect the contours (see Appendix B for details).

The Rényi entropy can then be represented as a path integral over just the output Hilbert space on the Keldysh contour $\mathcal{C}$ defined by Fig. 3(e). We parameterize this contour by the variable $s$, as shown in Fig. 3(e), taking the convention that it increases clockwise around each replica. This fixes the contour ordering of the fermions. Then, defining $N$ Majorana fields $\chi_i(s)$ in the output Hilbert space for $s \in [0, 4t + 2\epsilon) = [0, 4s_0)$, we have that the Rényi entropy is given by

$$S^{(2)} = -\log\left[\text{Tr}_{X_1 \cup Y_2}\left[\rho^2_{X_1 \cup Y_2}\right]\right] = -\log\left[\frac{\mathcal{Z}_2^{(X_1 \cup Y_2)}}{\mathcal{Z}(\epsilon)^2}\right], \tag{100}$$

where

$$\mathcal{Z}_2^{(X_1 \cup Y_2)} = \int \mathcal{D}\chi_i \, e^{-I_{\mathcal{C}}[\chi_i]}. \tag{101}$$

The action is given by

$$I_{\mathcal{C}} = \int_{\mathcal{C}} ds \left[\frac{1}{2}\sum_i \chi_i \partial_s \chi_i + f(s)H\right], \tag{102}$$

where $H$ is the Hamiltonian of Eq. (75). Here the factor $f(s) = 1, i, -i$ when $s$ is at a point on the contour describing imaginary time evolution, forward real time evolution, or backward real time evolution, respectively. Note that, in this language, the $R_4$ fermions have replica diagonal boundary conditions, while the contours for the $R_{1,2}$ fermions have a single twist operator inserted, and the $R_3$ fermions have two twist operator insertions.

Since we are dealing with a system with quenched disorder, we are really interested in computing quantities averaged over all disorder realizations of the couplings, $J_{i_1 \dots i_q}$:

$$\overline{S^{(2)}_{X_1 \cup Y_2}} = \overline{\log\left[\text{Tr}_{X_1 \cup Y_2}\left[\rho^2_{X_1 \cup Y_2}\right]\right]}, \tag{103}$$

where the overbar indicates a disorder average. Such a calculation can be performed via another replica trick. However, all disorder replica off-diagonal solutions will give subleading corrections in $1/N$, and so can be ignored in the large-$N$ limit. That is to say, as is standard in large-$N$ studies of SYK models, we can assume a disorder replica diagonal solution and approximate

$$\overline{\log\left[\text{Tr}_{X_1 \cup Y_2}\left[\rho^2_{X_1 \cup Y_2}\right]\right]} \approx \log\left[\overline{\text{Tr}_{X_1 \cup Y_2}\left[\rho^2_{X_1 \cup Y_2}\right]}\right]. \tag{104}$$

Hence, at the level of the path integral, we can directly average over the disordered couplings to obtain

$$I_{\mathcal{C}} = \frac{1}{2}\int_{\mathcal{C}} ds \sum_i \chi_i \partial_s \chi_i - \frac{NJ^2}{8}\int_{\mathcal{C}} ds \int_{\mathcal{C}} ds' F(s,s')\left(\frac{1}{N}\sum_i \chi_i(s)\chi_i(s')\right)^4, \qquad (105)$$

where $F(s,s') = f(s)f(s')$. As is standard in the treatment of SYK models, we introduce the bilocal Lagrange multiplier fields $\Sigma_i(s_1,s_2)$, which will correspond to the self-energies of the Majorana fermions, to enforce the constraint

$$G_i(s,s') = \frac{1}{M_i}\sum_{j\in R_i}\chi_j(s)\chi_j(s'). \qquad (106)$$

On-shell, $G_i(s_1,s_2)$ are the Green functions for the Majoranas. Note that we must introduce *different* Green functions for each of the regions $R_i$, as the fermions contained therein satisfy distinct boundary conditions. We denote by $M_i$ the number of fermions in $R_i$ and also define $\lambda_i = M_i/N$. On integrating out the fermions, we obtain the effective action

$$\frac{I_{\mathcal{C}}}{N} = -\frac{1}{2}\sum_{i=1}^4 \lambda_i \log\det\left[\partial_s - \Sigma_i\right] + \frac{1}{2}\int_{\mathcal{C}} dsds'\left[\sum_{i=1}^4 \lambda_i G_i(s,s')\Sigma_i(s,s')\right.$$
$$\left. -\frac{J^2}{4}F(s,s')\left(\sum_{i=1}^4 \lambda_i G_i(s,s')\right)^4\right], \qquad (107)$$

where the subscript on the derivative reminds us that we must apply the appropriate boundary conditions for the fermions in each region $R_i$. The saddle-point, or Schwinger-Dyson, equations for this action are given by

$$G_i = \left(\partial_s - \Sigma_i\right)^{-1}, \quad \Sigma_i(s,s') = J^2 F(s,s')\left[\sum_{j=1}^4 \lambda_j G_j(s,s')\right]^3 \equiv \Sigma(s,s'). \qquad (108)$$

Note that the self-energies for the fermions in each region $R_i$ are equal to one another. Since the action is proportional to $N$, the path integral in the large-$N$ limit is simply equal to the value of the integrand evaluated at the saddle-point. Explicitly, on-shell, we have that

$$\left.\frac{I_{\mathcal{C}}}{N}\right|_{\text{o.s.}} = \sum_{i=1}^4 \frac{\lambda_i}{2}\log\det(G_i) + \sum_{i=1}^4 \frac{\lambda_i}{2}\log\det\left(\partial_s\right) + \frac{1}{2}(\lambda_1 + \lambda_2 + 2\lambda_3 + 2\lambda_4)\log 2$$
$$+ \frac{3}{8}\int_{\mathcal{C}} ds \int_{\mathcal{C}} ds' \sum_{i=1}^4 \lambda_i G_i(s,s')\Sigma(s,s'), \qquad (109)$$

where we have introduced terms in the first line to regularize the fermion determinant, ensuring that we get the free-fermion result in the non-interacting limit of $J = 0$. Thus, computing the Rényi entropy reduces to solving the Schwinger-Dyson equations and substituting the results into Eq. (109). The thermal partition function is obtained in an analogous manner. These equations can be discretized and solved numerically using a straightforward self-consistent iterative procedure, the details of which are relegated to Appendix B.

Having established how to compute the required second Rényi entropies, we can now proceed to an analysis of the Rényi TOMI. First, we note that we can safely set the the regulator $\epsilon$ to zero since we are not dealing with a continuum theory, and so we do not expect any UV divergences. In this infinite temperature limit, the expression for the Rényi TOMI reduces to

$$I_3^{(2)}(A;C,D) = \frac{N}{2}\log 2 - S_{A\cup C}^{(2)} - S_{A\cup D}^{(2)}. \qquad (110)$$

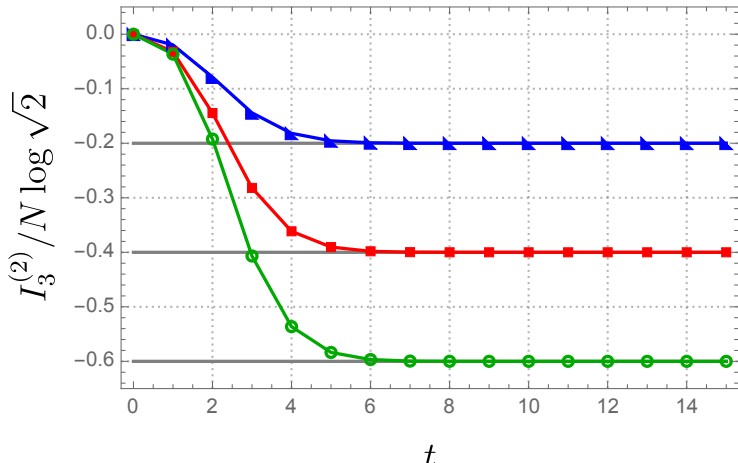

Figure 4: We show the large-$N$ result for $I_3^{(2)}(A; C, D)$ in the Majorana SYK model with $\lambda := N_A/N = 0.1$ (blue triangles), 0.2 (red squares), and 0.3 (green circles), where $N_A$ and $N$ are the numbers of fermions in $A$ and in the full system, respectively. The solid gray lines indicate the minimum possible value of $I_3^{(2)}$ for the given value of $\lambda$ and are clearly saturated at late times.

We will further specialize to the case where the output subspaces $C$ and $D$ are of equal size and disjoint, so that $N_C = N_D = N/2$, while $A \subset C$, so that $N_A < N/2$. In this case, the TOMI satisfies the bound $I_3^{(2)}(A; C, D) \geq -2N_A \log \sqrt{2}$, where $N_A \log \sqrt{2}$ is the $A$ region Hilbert space dimension, since each pair of Majorana fermions defines a qubit.

In the following numerical computation of the TOMI, we fix $\lambda = N_A/N$, where $N_A$ is the number of Majorana fermions in $A$ and $N$ is the total number of fermions. In the notation used in the path integral setup above, the non-trivial entropies are computed using the following configurations of the $\lambda_i$'s:

$$S_{A\cup C}^{(2)} : \lambda_1 = 0, \ \lambda_2 = \frac{1}{2} - \lambda, \ \lambda_3 = \lambda, \ \lambda_4 = \frac{1}{2}, \tag{111}$$

$$S_{A\cup D}^{(2)} : \lambda_1 = \lambda, \ \lambda_2 = \frac{1}{2}, \ \lambda_3 = 0, \ \lambda_4 = \frac{1}{2} - \lambda. \tag{112}$$

Our numerical results for $I_3^{(2)}(A; C, D)$ are depicted in Fig. 4. For all values of $\lambda$, the TOMI decreases monotonically before essentially saturating in a time independent of $\lambda$ to the value $-2N_A \log \sqrt{2}$, indicated by the solid gray lines. We thus find that the SYK model saturates the bound on the Rényi TOMI. This constitutes the first main result of this section. Indeed, we may therefore conclude that the SYK model (in the large-$N$ limit) exhibits stronger information scrambling than holographic CFTs. Conversely, by this measure, the large-$N$ SYK model is as strong a scrambler as the Brownian SYK model [65] and random unitary circuit models with large onsite Hilbert space dimensions [32], which are non-Hamiltonian systems. Hence, we see that conservation of energy is not by itself sufficient to suppress information scrambling.

How does the presence of additional conserved quantities modify this result? To answer this, we turn to the cSYK model defined above. The computation of the TOMI for the cSYK model in the $N \to \infty$ limit proceeds in exact analogy with that of the original SYK model, and so we omit the details. We need to compute

$$I_C = \int_C ds \left[ \sum_i c_i^\dagger \partial_s c_i + f(s) H \right], \tag{113}$$

where $\mathcal{C}$ is again the contour in Fig. 3(e) and $H$ is the complex SYK Hamiltonian. We will focus on the particle-hole symmetric point $\mu = 0$, as this is the case in which there are symmetry protected degeneracies in the energy spectrum of the finite-$N$ problem to be discussed later. We again let $M_i$ represent the number of complex fermions in subsystem $R_i$ and set $\lambda_i = M_i/N$. On performing the disorder average, we obtain the effective action

$$
\begin{aligned}
\frac{I_{\mathcal{C}}}{N} = &-\sum_{i=1}^{4} \lambda_i \log \det\left[\partial_{\substack{s\\R_i}} - \Sigma_i\right] - \int_{\mathcal{C}} ds \int_{\mathcal{C}} ds' \left[\sum_{i=1}^{4} \lambda_i G_i(s,s')\Sigma_i(s',s)\right. \\
&\left. + \frac{J^2}{4} F(s,s')\left[\sum_{i=1}^{4} \lambda_i G_i(s,s')\right]^2 \left[\sum_{i=1}^{4} \lambda_i G_i(s',s)\right]^2\right],
\end{aligned}
\tag{114}
$$

where

$$
G_i(s,s') = \frac{1}{M_i} \sum_{j \in R_i} c_j(s)c_j^\dagger(s').
\tag{115}
$$

As before, the $R_i$ in the derivative within the determinant is included to remind us that we must impose the appropriate boundary conditions for the fermions in each subsystem. The Schwinger-Dyson equations are given by

$$
G_i = \left(\partial_{\substack{s\\R_i}} - \Sigma_i\right)^{-1}, \quad \Sigma_i(s,s') = -J^2 F(s,s')\tilde{G}(s,s')^2 \tilde{G}(s',s),
\tag{116}
$$

where $\tilde{G} = \sum_i \lambda_i G_i$. Now, let us make use of the fact that we have fixed $\mu = 0$. The particle-hole symmetry generated by Eq. (90) implies that the Green functions satisfy $G_i(s,s') = -G_i(s',s)$. So, the equation for the self-energy becomes

$$
\Sigma_i(s,s') = J^2 F(s,s')\left[\sum_{i=1}^{4} \lambda_i G_i(s,s')\right]^3.
\tag{117}
$$

We see that the Schwinger-Dyson equations reduce to those for that of the standard SYK model while the on-shell action is twice that of the standard SYK model. The Green functions are thus trivially the same as those for the standard SYK model, while the TOMI saturates to twice the value of that for the standard SYK model. We can then immediately conclude that, at the particle-hole symmmetric point, the complex SYK model saturates the bound for the second Rényi TOMI. Hence, at least in the $N \to \infty$ limit, degeneracies protected by a single $U(1)$ conserved charge do not appear to inhibit information scrambling.

Before moving on to the analysis of these models for small values of $N$, a comparison with our results for holographic CFTs is in order. We found that the TOMI in holographic CFTs without an extended symmetry algebra saturates at a value parametrically smaller than the bound, while adding in a single $U(1)$ charge further reduces the saturation value by an $O(1)$ correction. As we argued above, this is because the Virasoro algebra results in degeneracies which grow *exponentially* in the level within each Verma module. In contrast, both the Majorana and cSYK models have at most double degeneracies protected by symmetries, and so by the general picture we have developed, one would not expect to see a suppression of the TOMI in the thermodynamic limit. We do, however, expect the effects of these degeneracies to be manifest in the finite-$N$ case, and it is this scenario to which we turn next.

### 3.2.2 Finite-$N$

We now simulate finite SYK systems for both Majorana and complex fermions. We also vary the number of fermions to see the impact of degeneracies. Representative examples are shown

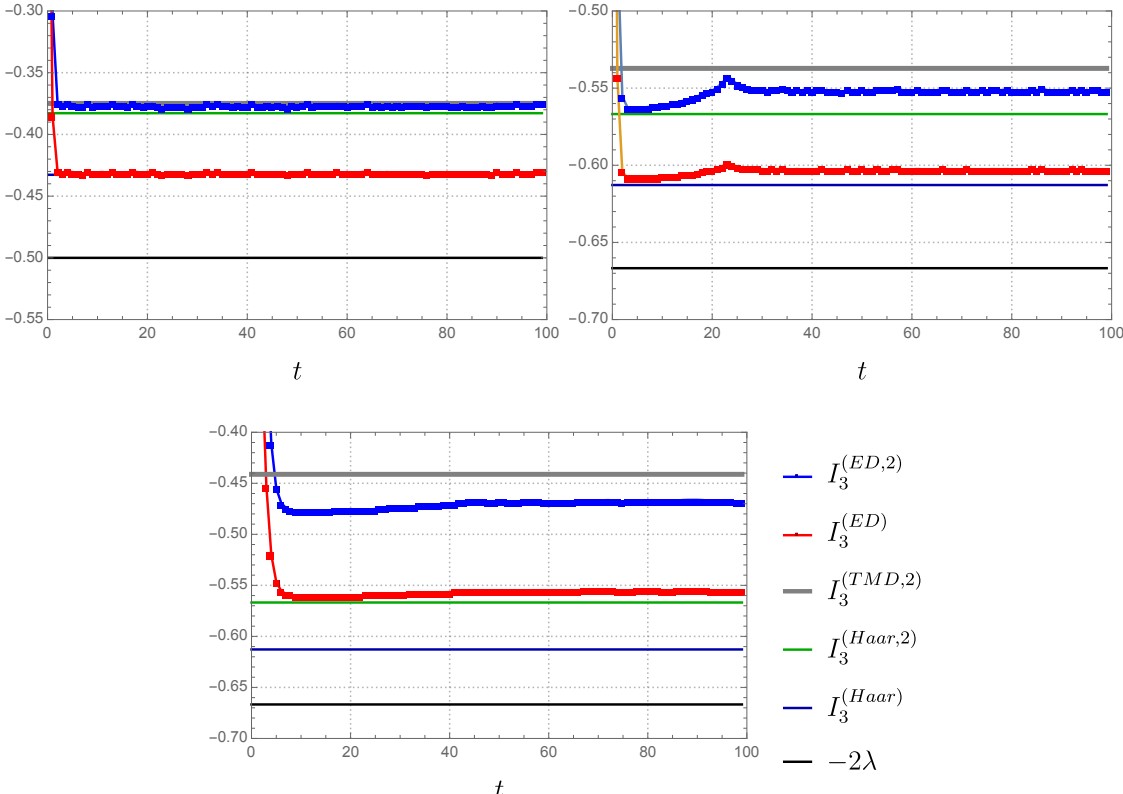

Figure 5: Plots of $I_3(A; C, D)$ (red squares) and $I_3^{(2)}(A; C, D)$ (blue squares) for the SYK models with different system sizes. Top Left: Majorana SYK with $N = 8$, where the energy spectrum has no degeneracy. $N_A = 2, N_C = N_D = 4$. Note that $I_3^{(TMD,2)}$ and the actual numerical data $I_3^{(ED,2)}$ are barely distinguishable. Top Right: Majorana SYK with $N = 12$, where the energy spectrum is doubly degenerate everywhere. $N_A = 4, N_C = N_D = 6$. Bottom: complex SYK with $N = 6$, where the energy spectrum is doubly degenerate everywhere. $N_A = 2, N_C = N_D = 3$. In every plot, each data point (squares), as well as the result from equilibrium approximation using TMD state (gray line), are obtained from averaging over 1000 disorder realizations.

in Fig. 5 with comparisons to the equilibrium approximation (with finite-size correction), Haar random unitaries, and the lower bound (6) plotted for comparison. It is immediately clear that (6) will not be saturated for finite-size systems even though it was saturated in the large-$N$ limit. More interestingly, we find the equilibrium approximation to provide a significantly better quantitative prediction for the late-time behavior than a Haar random matrix. In particular, when there are degeneracies in the spectrum, the equilibrium approximation prediction has noticeably smaller magnitude than the Haar random value due to its sensitivity to degeneracies. For the non-degenerate SYK model, the equilibrium approximation value and Haar value are very close but the equilibrium approximation is significantly more precise.

Having presented the finite-$N$ results above, we now discuss the issue regarding the finite-size correction of equilibrium approximation. In order to compare equilibrium approximation with finite-$N$ numerics, we must reconsider subleading corrections that were irrelevant in the thermodynamic limit. We do this explicitly for the second Rényi entropy where it is not too tedious to explicitly write out all subleading corrections. The first correction comes from the fact that we assumed there was a single dominant term in the sum over permutation, leading

to (18). In reality, we must sum over all terms. For the second Rényi entropy, there are only two terms because $|\mathcal{S}_2| = 2$, leading to

$$S_A^{(2)} = -\log\left[e^{-S_{A,eq}^{(2)}} + e^{-S_{\bar{A},eq}^{(2)}}\right].\tag{118}$$

This is the standard form of an expression describing the crossover between two large-$N$ saddles e.g. the free energy in the Hawking-Page transition.

The other important correction at finite-$N$ is from the approximation of the metric (14). Including subleading orders, the metric is not diagonal so we must consider the double sum over the symmetric group

$$
\begin{aligned}
\mathcal{Z}_2^{(A)} &= \frac{1}{Z_2^2} \sum_{\sigma,\tau \in \mathcal{S}_2} g^{\tau\sigma} \langle \eta_A \otimes e_B | \mathcal{I}_\alpha, \tau \rangle \langle \mathcal{I}_\alpha, \sigma | \rho_0, e \rangle \\
&= \frac{Z_1^2}{Z_2^2}\Big[ e^{-S_{A,eq}^{(2)}} \big( g^{ee} (\mathrm{Tr}[\rho_0 \mathcal{I}_\alpha])^2 + g^{e\eta} \mathrm{Tr}[(\rho_0 \mathcal{I}_\alpha)^2]\big) \\
&\qquad\qquad + e^{-S_{\bar{A},eq}^{(2)}} \big( g^{\eta\eta} (\mathrm{Tr}[\rho_0 \mathcal{I}_\alpha])^2 + g^{\eta e} \mathrm{Tr}[(\rho_0 \mathcal{I}_\alpha)^2]\big)\Big].
\end{aligned}
\tag{119}
$$

One can explicitly evaluate these terms for a given system. For SYK, we only consider the case where the spectrum is non-degenerate or doubly degenerate. In the case where the entire spectrum is $k^{th}$ degenerate, we have

$$g_{ee} = g_{\eta\eta} = 1, \quad g_{e\eta} = g_{\eta e} = \frac{1}{d}, \quad \mathrm{Tr}[(\rho_0 \mathcal{I}_\alpha)^n] = k^n, \quad Z_n = k^{n-1}d,\tag{120}$$

where $d$ is the Hilbert space dimension of one of the sides. The inverse metrics are then

$$g^{ee} = g^{\eta\eta} = \frac{d^2}{d^2 - 1}, \quad g^{e\eta} = g^{\eta e} = -\frac{d}{d^2 - 1},\tag{121}$$

leading to

$$\mathcal{Z}_2^{(A)} = \frac{d}{d+1}\left(e^{-S_{A,eq}^{(2)}} + e^{-S_{\bar{A},eq}^{(2)}}\right),\tag{122}$$

so the Rényi entropy of a subregion $A$ becomes

$$S_A^{(2)} = -\log\left[e^{-S_{A,eq}^{(2)}} + e^{-S_{\bar{A},eq}^{(2)}}\right] - \log\frac{d}{d+1}.\tag{123}$$

For our setup of the SYK model, the targeted region and its spatial complement are $A \cup C$ and $B \cup D$ respectively, and the equilibrium ensemble at late times can be constructed using the TMD state. Therefore, the Rényi entropy of $A \cup C$ computed from equilibrium approximation with finite-size correction is

$$S_{A\cup C}^{(2)} = -\log\left[e^{-S_{A\cup C}^{(2)}(\rho_{TMD})} + e^{-S_{B\cup D}^{(2)}(\rho_{TMD})}\right] - \log\frac{d}{d+1}.\tag{124}$$

There are also corrections coming from time-dependent terms that were dropped when taking the equilibrium approximation. These fluctuations may be the same order as the other subleading terms, but are expected to average to zero. For SYK, this averaging is done over the ensemble of couplings.

# 4 Connection to Operator Growth

It is always useful to approach the same physics problem from multiple perspectives, patching them together to generate a more complete understanding. In this section, we approach operator entanglement from the perspective of the growth of local operators under time evolution. Operator growth, especially in the form of the OTOC, has been a primary focus in the emerging field of many-body quantum chaos so it is important to make this connection.

Inspired by Refs. [76,77], in this section, we find a clean interpretation of operator mutual information in the language of operator growth and delocalization: The bipartite operator mutual information is the number of basis operators initially localized in region $A$ that are transported to region $C$ under time evolution. We are then led to understand the non-saturation of the lower bound for tripartite operator mutual information as counting the number of operators that are not fully delocalized under time evolution.

It is instructive to represent the density matrix in a basis of operators which we refer to as the *operator-gas formalism*

$$\rho = \sum_\alpha a_\alpha \mathcal{O}_\alpha. \tag{125}$$

We take the basis operators (barring the identity) to be traceless and orthogonal[12]

$$\langle \mathcal{O}_\alpha^\dagger \mathcal{O}_\beta \rangle = q^L \delta_{\alpha\beta}, \tag{126}$$

where $q$ is the dimension of the local Hilbert space and $L$ is the number of lattice sites. The evolution of the density matrix can then be understood as the evolution of local operators

$$\rho(t) = \sum_\alpha a_\alpha \mathcal{O}_\alpha(t). \tag{127}$$

In chaotic systems, initially localized operators have the tendency to grow in "size," eventually gaining non-trivial support on the entire system. This can be thought of as a signature (or definition) of operator scrambling. The time-evolved operators can once again be expanded in terms of the basis operators

$$\mathcal{O}_\alpha(t) = \sum_\beta c_\alpha^\beta(t) \mathcal{O}_\beta, \quad \sum_\beta \left| c_\alpha^\beta(t) \right|^2 = 1. \tag{128}$$

The constraint on the expansion coefficients, $c_\alpha^\beta(t)$, is a consequence of unitarity. These coefficients are the only theory-dependent inputs. They also have a natural interpretation; $\left| c_\alpha^\beta(t) \right|^2$ represents the probability of the basis operator $\mathcal{O}_\alpha$ to evolve into the basis operator $\mathcal{O}_\beta$ at time $t$.

To apply the operator gas formalism to operator entanglement, we note that the infinite temperature thermofield double state has a simple expansion

$$\rho_{TFD} = \frac{1}{q^{2L}} \sum_{\alpha=1}^{q^{2L}} \mathcal{O}_\alpha^{(1)} \otimes \mathcal{O}_\alpha^{(2)\dagger}, \tag{129}$$

where the superscripts denote the input and output Hilbert space respectively. The sum is over all basis operators on one copy of the Hilbert space and we do not impose a regulator because

---

[12]An example of such a basis is the clock and shift matrices that generalize the Pauli group.

we are working with a finite-dimensional Hilbert space. The state corresponding to the unitary operator then requires us to time evolve only the second copy

$$\rho^{(U)}(t) = \frac{1}{q^{2L}} \sum_{\alpha=1}^{q^{2L}} \mathcal{O}_\alpha^{(1)} \otimes \mathcal{O}_\alpha^{(2)\dagger}(t). \tag{130}$$

The reduced density matrix on $A \cup C$ is

$$\rho_{A\cup C}^{(U)}(t) = \frac{1}{q^{2L}} \sum_{\alpha}^{q^{2L}} \mathrm{Tr}_B\left[\mathcal{O}_\alpha^{(1)}\right] \otimes \mathrm{Tr}_D\left[\mathcal{O}_\alpha^{(2)\dagger}(t)\right]. \tag{131}$$

First, we expand the second operator in terms of basis operators

$$\rho_{A\cup C}^{(U)}(t) = \frac{1}{q^{2L}} \sum_{\alpha,\beta}^{q^{2L}} c_\alpha^{\beta*}(t) \mathrm{Tr}_B\left[\mathcal{O}_\alpha^{(1)}\right] \otimes \mathrm{Tr}_D\left[\mathcal{O}_\beta^{(2)\dagger}\right]. \tag{132}$$

Because of our choice of basis, the partial trace is simple. Every operator that has nontrivial support on $B \cup D$ does not contribute to the sum due to the tracelessness condition. The operators that are trivial on $\bar{A}/\bar{B}$ give factors of $q^{|B|+|D|}$, where $|\cdot|$ denotes the number of sites in the region. Thus, we have

$$\rho_{A\cup C}^{(U)}(t) = \frac{1}{q^{|A|+|C|}} \sum_{\alpha\in A, \beta\in C}^{q^{2|A|},q^{2|C|}} c_\alpha^{\beta*}(t) \mathcal{O}_\alpha^{(1A)} \otimes \mathcal{O}_\beta^{(2C)\dagger}, \tag{133}$$

where the sums are only over operators that are trivial outside of $A$ and $C$. The new superscripts on the basis operators denote that the normalization has changed to the dimension of the sub-Hilbert space. We frequently drop subscripts to simplify expressions. We can then compute the second Rényi entropy

$$e^{-S_{A\cup C}^{(2)}(t)} := \mathrm{Tr}\left[\rho_{A\cup C}^{(U)}(t)\right]^2 = \frac{1}{q^{2|A|+2|C|}} \sum_{\alpha,\alpha'\in A,\beta,\beta'\in C}^{q^{2|A|},q^{2|C|}} c_\alpha^{\beta*}(t) c_{\alpha'}^{\beta'}(t) \mathrm{Tr}\left[\mathcal{O}_\alpha \mathcal{O}_{\alpha'}^\dagger\right] \otimes \mathrm{Tr}\left[\mathcal{O}_\beta^\dagger \mathcal{O}_{\beta'}\right]. \tag{134}$$

Using the orthogonality of the basis operators, we find

$$S_{A\cup C}^{(2)}(t) = (|A|+|C|)\log q - \log N(A:C,t), \tag{135}$$

where we have defined the operator counting function

$$N(A:C,t) := \sum_{\alpha\in A}^{q^{2|A|}} \sum_{\beta\in C}^{q^{2|C|}} |c_\alpha^\beta(t)|^2. \tag{136}$$

This function has the interpretation as the expectation value of the number of operators that start in $A$ that evolve to operators in region $C$. The entropies of $A$ and $C$ individually are time-independent and maximal

$$S_A^{(2)} = |A|\log q, \quad S_C^{(2)} = |C|\log q, \tag{137}$$

so the operator mutual information is solely determined by the counting function

$$I^{(2)}(A,C) = \log N(A:C,t). \tag{138}$$

This formula is worth unpacking. First, it is important to note that this formula is completely general; we have not made any assumption about the structure of the $c_\alpha^\beta$'s. This means that this formula applies equally well to chaotic and integrable quantum systems and in any number of spatial dimension. It is an explicit connection between information scrambling attributed to the operator entanglement and operator scrambling attributed to operator growth and OTOCs. Eq. (138) states that the operator entanglement counts the number of operators starting in region $A$ in the input Hilbert subsystem that are transported to region $B$ in the output subsystem. Any operators that are delocalized under time evolution, as is generically expected in chaotic systems, will not contribute to the operator entanglement because they will have support in $D$. This is why the operator entanglement asymptotes to zero at late times in chaotic systems.

Some further intuition for (138) can be built by performing consistency checks. First, note that the counting function is always bounded below by unity because the identity operator always contributes to the sum. This is critical for the operator mutual information to be positive. Now, consider $A$ and $C$ to be symmetric intervals. At $t = 0$, the operators have not yet evolved, so we simply have $|c_\alpha^\beta|^2 = \delta_\alpha^\beta$. This truncates the double sum to a single sum and we obtain

$$N(A : C, t = 0) = q^{2|A|} = q^{2|C|},$$
$$\Rightarrow I^{(2)}(A, C, t = 0) = 2|A| \log q = 2|C| \log q, \tag{139}$$

as expected because we start with a collection of $|A|$ generalized Bell pairs. Next, consider disjoint regions at $t = 0$. In this case, by definition, only the identity operator contributes and we obtain a trivial mutual information, as expected. Finally, take $C$ to be the entire output. Because of the identity $\sum_\beta \left| c_\alpha^\beta(t) \right|^2 = 1$, we have maximal entanglement for all times

$$N(A : C, t) = q^{2|A|} \Rightarrow I^{(2)}(A, C, t) = 2|A| \log q. \tag{140}$$

With (138) in hand, we can express the tripartite operator mutual information in terms of the counting function

$$I_3^{(2)}(A; C, D) = \log \left[ \frac{N(A : C, t) N(A : D, t)}{N(A : C \cup D, t)} \right]. \tag{141}$$

This means that the tripartite operator mutual information counts the number of operators that are localized in $C$ and $D$. Usually, we take $C \cup D$ to be the entire output Hilbert space, in which case

$$I_3^{(2)}(A; C, D) = \log \left[ N(A : C, t) N(A : D, t) \right] - 2|A| \log q. \tag{142}$$

Note that the second term represents the fundamental lower bound that signals maximal scrambling (6). If the first term is trivial, the lower bound is saturated. The first term being trivial means that there are no operators that remain localized in either $C$ or $D$ at late times i.e. all operators are fully delocalized. Thus, maximal scrambling in the sense of operator entanglement is equivalent to the statement of all operators growing to gain support on the entire system at late times. Any system that does not saturate the lower bound has operators that remain localized at late times.

## 4.1  Implications for Holographic CFTs

While we found SYK models (at large-$N$) saturate the lower bound, implying all operators are fully delocalized,[13] the same was not true for 2D holographic CFTs. Here, we interpret this

---

[13]Due to the infinite number of degrees of freedom, the more precise statement is that a measure one subset of basis operators are fully delocalized.

result from the operator growth perspective. First, we consider the case without the $\mathfrak{u}_k(1)$ symmetry. The effective number of basis operators in a subsystem of length $l$ is $e^{\frac{\pi c L_A}{4\epsilon_{UV}}}$. This can be read off from $I^{(2)}(A, C)$. From (65), we find the operator counting functions

$$N(A : C) = N(A : D) = e^{\frac{\pi(c+2)L_A}{24\epsilon_{UV}}}. \tag{143}$$

Thus, an exponentially large number of basis operators remain localized, though this is an exponentially small percentage of the total number of operators. These relatively few operators play an important role in entanglement dynamics. For instance, they are responsible for the apparent "quasi-particle dip" found in Ref. [23]. From this perspective, we should not consider holographic CFTs to be maximally scrambling as a certain portion of the operators are not scrambled. These coherent operators are likely a consequence of the Virasoro symmetry and related to the quantum KdV currents, the chiral conserved operators present in all 2D CFTs [78, 79]. Currently, we do not know how to make the mapping between these operators and the KdV currents precise.

Next, consider holographic CFTs with the $\mathfrak{u}_k(1)$ current algebra. The operator counting function becomes

$$N(A : C) = N(A : D) = e^{\frac{\pi(c+4)L_A}{24\epsilon_{UV}}}. \tag{144}$$

We therefore understand the role of conserved quantities as placing constraints on the dynamics such that certain operators do not delocalize, leading to a suppression of quantum information scrambling as measured by the TOMI.

## 4.2 OTOCs

We have frequently referenced OTOCs as a measure of operator growth and scrambling. It is therefore useful to understand what OTOCs look like in the operator gas picture. Consider the OTOC of two basis (non-identity) operators

$$C_{\alpha\beta}(t) = \langle \mathcal{O}_{\alpha_i}^\dagger \mathcal{O}_{\beta_j}^\dagger(t) \mathcal{O}_{\alpha_i} \mathcal{O}_{\beta_j}(t) \rangle. \tag{145}$$

The expectation value is evaluated at infinite temperature, so we have

$$\begin{aligned} C_{\alpha\beta}(t) &= \frac{1}{q^L} \mathrm{Tr}\left[ \mathcal{O}_{\alpha_i}^\dagger \mathcal{O}_{\beta_j}^\dagger(t) \mathcal{O}_{\alpha_i} \mathcal{O}_{\beta_j}(t) \right] \\ &= \frac{1}{q^L} \sum_{\gamma,\delta} c_{\beta_j}^{\gamma*}(t) c_{\beta_j}^\delta(t) \mathrm{Tr}\left[ \mathcal{O}_{\alpha_i}^\dagger \mathcal{O}_\gamma^\dagger \mathcal{O}_{\alpha_i} \mathcal{O}_\delta \right]. \end{aligned} \tag{146}$$

Here, $\mathcal{O}_{\alpha_i}$ represents $\mathcal{O}_\alpha$ on site $i$ tensored with identities on all other sites. Clearly, the trace is always zero when $\mathcal{O}_\gamma$ and $\mathcal{O}_\delta$ are not equivalent away from site $i$, so the sum reduces to

$$C_{\alpha\beta}(t) = \sum_{\gamma=\cdots\otimes\mathbb{1}_i\otimes\ldots} \left| c_{\beta_j}^\gamma(t) \right|^2 + \frac{1}{q^L} \sum_{\gamma,\delta\neq\cdots\otimes\mathbb{1}_i\otimes\ldots} c_{\beta_j}^{\gamma*}(t) c_{\beta_j}^\delta(t) \mathrm{Tr}\left[ \mathcal{O}_{\alpha_i}^\dagger \mathcal{O}_\gamma^\dagger \mathcal{O}_{\alpha_i} \mathcal{O}_\delta \right]. \tag{147}$$

At early times, the second sum is trivial because the operators have not had time to reach site $i$. Thus, the OTOC is equal to unity at early times.

The trace involving four nontrivial operators can, in general, be complicated. For maximal clarity, we restrict to the basis of ($q = 2$) Pauli operators where we have the following well-known trace relation

$$\begin{aligned} \mathrm{Tr}\left[ \sigma_\alpha \sigma_\beta \sigma_\gamma \sigma_\mu \right] &= 2\left( \delta_{\alpha\beta}\delta_{\gamma\mu} + \delta_{\alpha\mu}\delta_{\beta\gamma} - \delta_{\alpha\gamma}\delta_{\beta\mu} \right) \\ &\quad + 4\left( \delta_{\alpha\gamma}\delta_{0\beta}\delta_{0\mu} + \delta_{\beta\mu}\delta_{0\alpha}\delta_{0\gamma} \right) - 8\delta_{0\alpha}\delta_{0\beta}\delta_{0\gamma}\delta_{0\mu} \\ &\quad + 2i \sum_{(\alpha\beta\gamma\mu)} \varepsilon_{0\alpha\beta\gamma}\delta_{0\mu}, \end{aligned} \tag{148}$$

where the index 0 represents the identity matrix, $\varepsilon$ is the generalized Levi-Civita tensor, and the sum is over cyclic permutations. For our purposes, $\alpha = \gamma \neq 0$, so there is a large simplification

$$\text{Tr}\left[\sigma_\alpha \sigma_\beta \sigma_\alpha \sigma_\mu\right] = 2\left(2\delta_{\alpha\beta}\delta_{\alpha\mu} - \delta_{\beta\mu}\right) + 4\delta_{0\beta}\delta_{0\mu}. \tag{149}$$

Plugging this into (147), we find

$$C_{\alpha\beta}(t) = \sum_{\gamma = \cdots \otimes \mathbb{1}_i \otimes \ldots} \left|c^\gamma_{\beta_j}(t)\right|^2 - \sum_{\gamma \neq \cdots \otimes \mathbb{1}_i \otimes \ldots} \left|c^\gamma_{\beta_j}(t)\right|^2 + 2\sum_{\gamma \neq \cdots \otimes \mathbb{1}_i \otimes \ldots} \left|c^{\alpha_i}_{\beta_j}(t)\right|^2, \tag{150}$$

where $\alpha|_i$ means that the operator, restricted to site $i$, is $\mathcal{O}_{\alpha_i}$. The various sums make this formula slightly confusing. However, it may simply be interpreted as the probability of the operator $\mathcal{O}_\beta$ localized at $j$ evolving to an operator that is either trivial or $\mathcal{O}_\alpha$ restricted to site $i$ minus the probability of it evolving to a different operator restricted to site $i$. Ergodicity implies that at late times $\mathcal{O}_\beta$ will have an equal probability of being any member of the Pauli group. This means that the OTOC is equal to 0, implying scrambling.

## 5 Discussion

We have initiated a study of the role of symmetry-enforced degeneracies in the energy spectra of Hamiltonian systems in inhibiting information scrambling, as measured by the tripartite operator mutual information (TOMI). Making use of the recently developed framework of equilibrated pure states, we demonstrated how degenerate energy levels result in off-diagonal correlations which survive dephasing in the long-time limit, leading to a non-saturation of the TOMI. Specifically, we illustrated that the long-time, coarse-grained features of a chaotic system quenched from the thermofield double state, as employed in the computation of operator entanglement, are well-described by the thermomixed double state (TMD) when only energy is conserved, or a charged TMD state – which supports off-diagonal elements reflecting residual quantum correlations – when additional symmetries are present. As case studies of this general picture, we investigated the Rényi TOMIs of holographic CFTs and Sachdev-Ye-Kitaev models, finding non-saturation in the presence of degeneracies and exceptional agreement with the saturation value predicted by the (charged) TMD state. We further connected our results on information scrambling to the picture of operator growth, demonstrating that a non-saturation of the TOMI necessarily implies the localization of some set of operators. Our work provides a new window into the role of symmetries and spectrum degeneracies in information scrambling in Hamiltonian systems and provides a foundation for further study of these systems, as we now describe.

Indeed, an important line of inquiry is to establish a more quantitative connection between the degree of degeneracy in the spectrum of a Hamiltonian system and the level of suppression of the long-time limit of the TOMI. In the cases studied in this paper, we found that the Rényi TOMI was not saturated in holographic CFTs but was saturated in the large-$N$ limit of SYK models. We attributed this difference to the fact that the former class of theories have an extensive number of degeneracies, while the SYK models (at a fixed value of $N$) have at most a double degeneracy, which should become unimportant in the large-$N$ limit. While this is certainly a reasonable heuristic, it would be desirable to make use of a quantitative measure of the level of degeneracy to establish a bound on the TOMI which makes this statement more precise. In Appendices C and D we made some preliminary remarks in this direction, noting that the average degeneracy in a spectrum can be extracted from the spectral form factor, while Pinsker's inequality can be used to bound entanglement in the charged TMD state. We also saw that a charged TMD state becomes increasingly distant from the TMD state as the

level of degeneracy increases. These provide some hints at a more quantitative theory of the suppression of information scrambling due to symmetry-enforced degeneracies.

Along this line, it is also important to note that our arguments from the equilibrated pure state picture rested on the claim that subsystem entanglement in the TMD state always "looks" thermal. Although we supported this statement with numerics and we have no physical reason to doubt its veracity, it is an important problem to establish this result analytically.

Looking further beyond, recent years have seen increasing interest in *symmetry-resolved* measures of entanglement [80–86]. For a system with some global symmetry, one can compute the contribution of each charge sector to the total, say, entanglement entropy of a given state. A natural extension of these prior works on symmetry-resolved state entanglement is to consider symmetry-resolved operator entanglement. Computations of such measures in chaotic systems should shed light on how different charge sectors contribute to information scrambling and how information is scrambled between these charge sectors. Such considerations are of particular interest in light of the connection between the TOMI and localization of operators established in Section 4 of this paper. Indeed, one may consider developing a symmetry-resolved operator gas picture of operator growth, which would account for how operators with a given charge with respect to some global symmetry grow into operators of other charges and should presumably be connected to a symmetry-resolved generalization of the TOMI. Fleshing out these connections will provide further insight into the role of symmetry in quantum chaos in Hamiltonian systems.

It has also recently been argued that chaotic systems exhibiting diffusive transport of a conserved quantity may exhibit *diffusive* growth of the Rényi entanglement entropy after a global quench [12–16] instead of the ballistic growth generically expected for non-integrable systems [87]. The computation of a symmetry-resolved TOMI in these systems, with its connection to operator growth, may assist in isolating the contributions to the entanglement growth from different classes of operators characterized by their charge under the appropriate symmetry.

Given the connection between operator entanglement and operator growth/localization, it is also natural to consider extensions of the present work to studies of systems with extensive numbers of *emergent* conserved quantities, namely many-body localized (MBL) phases. Such systems exhibit logarithmic growth of entanglement under global quenches [88,89]. It would be interesting to incorporate these nearly conserved quantities by extending the equilibrated pure state formalism by making the effective identity operator different at different time scales.

Finally, we note that because we have shown that our predictions hold for small quantum systems, they should be accessible in current experiments and noisy intermediate-scale quantum (NISQ) technology [90]. We believe that this is within reach as efficient experimental protocols have been constructed for both preparing thermofield double states [91–93] and evaluating Rényi entropies [94].

# Acknowledgments

We thank Hong Liu, Shinsei Ryu, and Shreya Vardhan for helpful discussions and comments. JKF is supported through a Simons Investigator Award to Shinsei Ryu from the Simons Foundation [Award Number: 566166]. RS acknowledges the support of the Natural Sciences and Engineering Research Council of Canada (NSERC) [funding reference number 6799-516762-2018]. This work was also supported in part by the US National Science Foundation through the NSF under grant No. DMR-1725401 at the University of Illinois (LN, RS). We use QUS-PIN for simulating finite-$N$ dynamics in Section 3 [95,96]. This work made use of the Illinois Campus Cluster, a computing resource that is operated by the Illinois Campus Cluster Program

(ICCP) in conjunction with the National Center for Supercomputing Applications (NCSA) and which is supported by funds from the University of Illinois at Urbana-Champaign.

# A  Operator Entanglement in Higher Dimensions

The case studies in the main text were limited to zero (SYK) and one (holographic CFTs) spatial dimensions. It is clearly desirable to study higher dimensional theories. We are, however, limited in our capacity to do so because general knowledge of analytic techniques for chaotic theories in higher dimensions is lacking.

Notable exceptions to the prior statement are higher dimensional conformal field theories that are holographically dual to semi-classical Einstein gravity theories in one higher dimension. While these CFTs are extremely complicated and chaotic quantum field theories, remarkably, entanglement entropy may be computed rather easily by appealing to the gravity theory. The Ryu-Takayanagi formula [43, 44] and its covariant generalization that was put forward by Hubeny, Rangamani, and Takayanagi [45] states the the entanglement entropy of a subregion of the CFT is equal to the area (in Planck units) of the extremal surface in the gravity theory, $\gamma_A$, that is anchored on $A$

$$S_{vN}(A) = \frac{\text{Area}(\gamma_A)}{4G_N}, \tag{151}$$

where $G_N$ is Newton's constant.

The thermofield double state of the CFT is dual to the eternal black hole in the gravity theory [97]. This is a geometry that has two asymptotically anti-de Sitter regions and is connected by a wormhole. To compute operator entanglement, we are thus instructed to determine the areas of surfaces in this geometry that are anchored on $A \cup B$, where $A$ and $B$ are arbitrary regions on opposite sides of the wormhole. Notably, the relevant surfaces probe the interior geometry of the black hole [98].

Though certainly tractable, we will not explicitly calculate the areas of these surfaces, instead appealing to a more universal framework that may be applicable to generic chaotic theories. In recent years, it has become more clear that a large class of theories undergo entanglement dynamics that can be geometrized in an analogous way to holographic theories. Rather than the extremal surface extending into the bulk of a higher dimensional gravitational theory, in the *membrane theory of entanglement dynamics*, one considers an extremal surface in the spacetime of the theory of interest, no holography needed. Just as in holography, the surface must be homologous to the subregion of interest (see Fig. 6). We call the cost functional the membrane tension, $\mathcal{E}(\bar{v})$. In a homogeneous system, this tension will only depend on its local "velocity." The entanglement entropy of a region is then given by

$$S_A^{(n)} = s_{eq}^{(n)} \int_{\mathcal{M}} d^{d-1}x \, \mathcal{E}(\bar{v}), \tag{152}$$

where $s_{eq}^{(n)}$ is the thermal Rényi entropy density. The membrane tension is the single dynamical input that specifies the theory. This new effective theory of entanglement dynamics was initially motivated by random unitary circuit models [46, 47] but has proven to be widely applicable to quantum chaotic systems, entanglement measures, and quench protocols [32, 36, 48–51, 99–101]. Connecting further with holographic entanglement entropy, it has been shown that (152) follows from (151) in holographic theories by a projection of the relevant portion of the extremal surface [50].

We now argue that, very generally, any theory that follows the membrane picture must saturate (6). For simplicity, we consider each region to be translationally invariant in all directions except the $x$ direction, where they are of linear sizes $L_A$ and $L_B$, thus reducing the

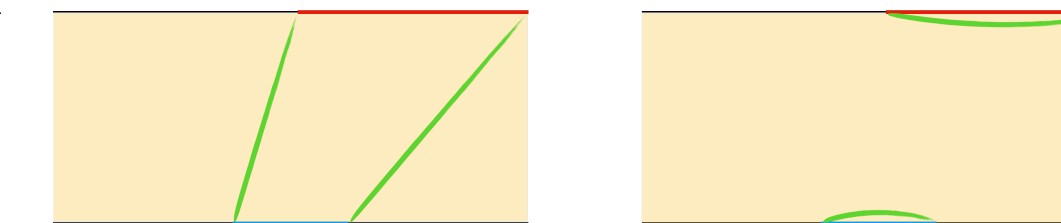

Figure 6: We show a projection of spacetime onto a single spatial dimension with time running vertically. To evaluate the Rényi entropies of $A$ (blue line located at $t = 0$) and $B$ (red line located at time $t$), we find the candidate membranes (green lines). At early times, the membrane in the left figure will be minimal. It is clearly time-dependent because as time progresses, $A$ and $B$ are brought further away from one another. At late times, the membranes in the right figure will be minimal. These are clearly time-independent and represent the thermal values.

computation to a 1+1D problem. There are two competing membranes for $S(A \cup B)$ (see Fig. 6). The time-dependent membrane stretches across from the input to output subsystems. Assuming a homogeneous Hamiltonian, the membrane, which is split into two disconnected pieces, will have constant velocities

$$S_{A \cup B}^{(n)} = s_{eq}^{(n)} \Sigma \left( \mathcal{E}(\bar{v}_1) + \mathcal{E}(\bar{v}_2) \right) t, \tag{153}$$

where $\Sigma$ is the area of the transverse directions. The time-independent membrane simply represents the thermal entropy assuming $A \cup B$ is less than half the total system size

$$S_{A \cup B}^{(n)} = s_{eq}^{(n)} \Sigma (L_A + L_B). \tag{154}$$

The individual entropies of $A$ and $B$ are constant, so the mutual information is

$$I^{(n)}(A, B) = s_{eq}^{(n)} \Sigma \max \left[ 0, L_A + L_B - \left( \mathcal{E}(\bar{v}_1) + \mathcal{E}(\bar{v}_2) \right) t \right]. \tag{155}$$

At late times, this means that the mutual information will always be zero, so the tripartite mutual information is

$$I_3^{(n)} = -2 s_{eq}^{(n)} \Sigma L_A, \tag{156}$$

which is the lower bound. When combined with the results from Sec. 3.1, this implies that 2D holographic CFTs cannot obey a membrane theory for Rényi entropies.

# B  Numerical Scheme for Large-$N$ SYK

In this appendix, we provide details on our numerical evaluation of the Rényi TOMI for the Majorana SYK model. The main point at issue is to properly account for the boundary conditions of the Majorana fermions in each of the $R_i$ regions. This is done by replacing the partial derivatives, $\partial_s \big|_{R_i}$, by the inverse free Green functions, $G_{0,i}^{-1}$ of the $R_i$ fermions since, by definition, they satisfy $\partial_s \big|_{R_i} G_{0,i}(s, s') = \delta(s - s')$.

It thus remains to solve a simple ordinary differential equation for the $G_{0,i}$. To do so, it is

convenient to define four species of fermions $\chi_i^\alpha$, $\alpha = 1, 2, 3, 4$ as

$$\chi_i(s) = \begin{cases} \chi_i^{\alpha=1}(s) & s \in [0, s_0) \\ \chi_i^{\alpha=2}(s - s_0) & s \in [s_0, 2s_0) \\ \chi_i^{\alpha=3}(s - 2s_0) & s \in [2s_0, 3s_0) \\ \chi_i^{\alpha=4}(s - 3s_0) & s \in [3s_0, 4s_0), \end{cases} \tag{157}$$

where we have set $s_0 = t + \epsilon/2$. The fermions $\chi_i^\alpha(s)$ are hence defined in the range $s \in [0, s_0)$. In Fig. 3(e) we have labeled the four contour segments by $\alpha = 1, 2, 3, 4$, with each corresponding to the overlaps appearing in Eq. (98):

$$\alpha = 1: \langle y\overline{y}' | U_1^\dagger | x\overline{x}' \rangle \quad \alpha = 2: \langle x\overline{x} | U_1 | y\overline{y} \rangle \tag{158}$$

$$\alpha = 3: \langle y'\overline{y} | U_1^\dagger | x'\overline{x} \rangle \quad \alpha = 4: \langle x'\overline{x}' | U_1 | y'\overline{y}' \rangle. \tag{159}$$

In our parameterization of the contour, the $\alpha = 1, 2, 3, 4$ segments correspond to the following ranges of $s$, respectively: $[0, s_0)$, $[s_0, 2s_0)$, $[2s_0, 3s_0)$, $[3s_0, 4s_0)$. The four $\chi_i^\alpha$ fermion species are thus defined, respectively, on each of the four Keldysh contour segments. The boundary conditions for the Majorana fermions in each spatial subregion $R_i$ of the output Hilbert space are then obtained by matching up the bras and kets in Eq. (98). Explicitly, we have

$$\chi_i^1(s_0^-) = \chi_i^2(0^+); \ \chi_i^2(s_0^-) = \chi_i^3(0^+); \ \chi_i^3(s_0^-) = \chi_i^4(0^+); \ \chi_i^4(s_0^-) = -\chi_i^1(0^+) \quad i \in R_1$$
$$\chi_i^1(s_0^-) = \chi_i^4(0^+); \ \chi_i^4(s_0^-) = -\chi_i^3(0^+); \ \chi_i^3(s_0^-) = -\chi_i^2(0^+); \ \chi_i^2(s_0^-) = -\chi_i^1(0^+) \quad i \in R_2$$
$$\chi_i^1(s_0^-) = \chi_i^4(0^+); \ \chi_i^4(s_0^-) = -\chi_i^1(0^+); \ \chi_i^2(s_0^-) = \chi_i^3(0^+); \ \chi_i^3(s_0^-) = -\chi_i^2(0^+) \quad i \in R_3$$
$$\chi_i^1(s_0^-) = \chi_i^2(0^+); \ \chi_i^2(s_0^-) = -\chi_i^1(0^+); \ \chi_i^3(s_0^-) = \chi_i^4(0^+); \ \chi_i^4(s_0^-) = -\chi_i^3(0^+) \quad i \in R_4. \tag{160}$$

The minus signs arise from taking into account the contour-ordering of the path integral defined by our parameterization of the contour in terms of $s$. The free Green functions for the $\chi_i^\alpha$ fermions satisfying these boundary conditions are given by

$$G_{0,i}^{\alpha\beta}(s, s') = \frac{1}{2} \delta^{\alpha\beta} \operatorname{sgn}(s - s') + A_i^{\alpha\beta}, \quad s \in [0, s_0), \tag{161}$$

where the $A_i^{\alpha\beta}$ are $s$-independent matrices:

$$A_1^{\alpha\beta} = \frac{1}{2} \begin{pmatrix} 0 & -1 & -1 & -1 \\ +1 & 0 & -1 & -1 \\ +1 & +1 & 0 & -1 \\ +1 & +1 & +1 & 0 \end{pmatrix}, \qquad A_2^{\alpha\beta} = \frac{1}{2} \begin{pmatrix} 0 & -1 & +1 & -1 \\ +1 & 0 & -1 & +1 \\ -1 & +1 & 0 & -1 \\ +1 & -1 & +1 & 0 \end{pmatrix}, \tag{162}$$

$$A_3^{\alpha\beta} = \frac{1}{2} \begin{pmatrix} 0 & 0 & 0 & -1 \\ 0 & 0 & -1 & 0 \\ 0 & +1 & 0 & 0 \\ +1 & 0 & 0 & 0 \end{pmatrix}, \qquad A_4^{\alpha\beta} = \frac{1}{2} \begin{pmatrix} 0 & -1 & 0 & 0 \\ +1 & 0 & 0 & 0 \\ 0 & 0 & 0 & -1 \\ 0 & 0 & +1 & 0 \end{pmatrix}. \tag{163}$$

From these expressions for $G_{0,i}^{\alpha\beta}(s, s')$, it is straightforward to obtain expressions for the $G_i(s, s')$. These explicit forms of the twisted Green functions allow us to numerically solve the Schwinger-Dyson equations and compute the action, as we now describe.

In order to numerically evaluate the path integral in the large-$N$ limit, we discretize the contour into $L$ points, turning the Green functions and self-energies into $L \times L$ matrices (since the boundary conditions on the contour explicitly break time translation, we cannot simplify the computation with fast Fourier transforms). In particular, we divide each real time interval

[e.g. $s \in [0,t]$] into $T$ points and each imaginary time interval [e.g. $s \in [t,t+\epsilon]$] into $B$ points, so that $L = 4T + 2B$. We then have $ds \to \Delta s_i$, where $\Delta s_i = t/T$ or $\Delta s_i = \epsilon/B$, depending on whether $s_i$ lies in a real time or imaginary time segment, respectively. Explicitly, our discretization prescription is as follows:

$$\delta(s-s')\partial_s \underset{R_i}{\to} \frac{1}{\Delta s_m \Delta s_n}(G_{0,i})^{-1}_{mn}, \tag{164}$$

$$G_i(s,s') \to (G_i)_{mn}, \tag{165}$$

$$\Sigma_i(s,s') \to \frac{1}{\Delta s_m \Delta s_n}(\Sigma_i)_{mn}, \tag{166}$$

$$F(s,s') \to F_{mn}, \tag{167}$$

where $m,n \in \{1,\ldots,L\}$ and $G_{0,i}$ is the free Green function for the Majorana fermions in region $R_i$, subject to the appropriate boundary conditions. Note that we must introduce a factor of $1/\Delta s_m \Delta s_n$ in the discretization of $\delta(s-s')\partial_s$ when replacing it with the free inverse Green function to ensure we have the correct units. The rescaling of the self-energies by the same factor is just done for convenience.

The discretized versions of the Schwinger-Dyson equations are then given by

$$(G_i)_{mn} = \left(G_{0,i}^{-1} - \Sigma_i\right)^{-1}_{mn}, \tag{168}$$

$$(\Sigma_i)_{mn} = J^2 F_{mn} \Delta s_m \Delta s_n \Big[\sum_{j=1}^{4} \lambda_j G_j\Big]^3_{mn} \equiv (\Sigma)_{mn}. \tag{169}$$

When evaluated on-shell, the action is given by

$$\begin{aligned}
\frac{I_{\mathcal{C}}}{N}\bigg|_{\text{on-shell}} &= -\sum_{i=1}^{4}\frac{\lambda_i}{2}\log\det\left(G_{0,i}^{-1} - \Sigma\right) + \frac{3}{8}\sum_{m,n}\sum_{i=1}^{4}\lambda_i(G_i)_{mn}(\Sigma)_{mn} \\
&\quad + \sum_{i=1}^{4}\frac{\lambda_i}{2}\log\det\left(G_{0,i}^{-1}\right) - \frac{1}{2}(\lambda_1 + \lambda_2 + 2\lambda_3 + 2\lambda_4)\log 2.
\end{aligned} \tag{170}$$

Note that we have regularized the action by adding in the terms in the second line to ensure we get the free fermion result when $J = 0$.

In order to solve the Schwinger-Dyson equations, we employ a standard iterative self-consistent weighted update procedure [61]. We take the free Green functions as our *ansätze*, so that in iteration $l = 1$, we have that $G_i^{(l=0)} = G_{0,i}$. We then compute the self-energy $[\Sigma]^{(l)}$ using the Schwinger-Dyson equations of Eq. (169) and perform a weighted update of the Green functions:

$$G_i^{(l+1)} = (1-x)G_i^{(l)} + x\left(G_{0,i}^{-1} - \Sigma^{(l)}\right)^{-1}, \tag{171}$$

where we take $x = 0.5$. The Green functions of the $l^{th}$ iteration are then used as input for the subsequent iteration. When the change in Green functions between iterations

$$\frac{1}{L^2}\sum_{i=1}^{4}\sum_{m,n}\lambda_i\left[(G_i^{(l)})_{m,n} - (G_i^{(l)})_{m,n}\right] \tag{172}$$

drops below a threshold of $\epsilon = 10^{-10}$ and the Schwinger-Dyson equations are satisfied to the same tolerance, we say that the iterative procedure has converged and use the Green functions to compute the action, Eq. (170). We compute the action for several values of $L$ using this procedure and then linearly extrapolate the results as functions of $1/L$ to obtain the action in the $L \to \infty$ limit.

## C  Entropy Bound

In this appendix, we make more mathematically precise the statement that off-diagonal elements of the density matrix in the energy eigenbasis cause the entropies to decrease. We invoke Pinsker's inequality which upper bounds the trace distance between density matrices using the relative entropy

$$D(\rho||\sigma) \geq \frac{1}{2}|\rho - \sigma|_1^2. \tag{173}$$

The definition of the relative entropy is

$$D(\rho||\sigma) := \mathrm{Tr}\rho \log \rho - \mathrm{Tr}\rho \log \sigma. \tag{174}$$

Taking $\sigma$ to be proportional to the identity operator

$$\sigma = \frac{\mathbb{1}}{d}, \tag{175}$$

where $d$ is the dimension of the Hilbert space, the relative entropy becomes equal to the difference between the entropy of $\rho$ and the maximal (thermal) entropy

$$D(\rho||\sigma) = \log d - S_{vN}(\rho) = \Delta S_{vN}(\rho, \sigma). \tag{176}$$

From Pinsker's inequality, we have that the difference between the TMD value and the true equilibrium density matrix entropy is *at least* equal to half the trace distance squared. The trace distance is zero if and only if $\rho = \sigma$, so because $\rho$ has off-diagonal elements $\Delta S_{vN}(\rho, \sigma)$ is strictly positive i.e. the entropy is strictly less than $\log d$. It would be interesting to find stronger bounds on $\Delta S_{vN}(\rho, \sigma)$.

## D  Average Degeneracies

Given that degeneracies play a key role in the suppression of information scrambling, we would like to develop a way to estimate the average number of degeneracies at a given energy scale $\beta$. First, consider the spectral form factor

$$g(\beta, t) := |\mathscr{Z}(\beta/2 + it)|^2 = \sum_{n,m} e^{-(\beta/2+it)E_n - (\beta/2-it)E_m}. \tag{177}$$

In chaotic theories, $g(\beta, t)$ has characteristic features that show the similarity of the energy spectrum to the spectra of random matrices. First, $g(\beta, t)$ decays as a power law as the quantity dephases. When $t$ becomes $O(e^{S/2})$, where $S$ is the entropy, the decay transitions to a linear increase, a signature of the long range eigenvalue repulsion of random matrices. At $t \sim O(e^S)$, $g(\beta, t)$ saturates to a (rapidly oscillating) plateau. It is this late time value that will be most important for us because

$$\lim_{t \to \infty} g(\beta, t) = \sum_n d(E_n)e^{-\beta E_n}, \tag{178}$$

where $d(E_n)$ is the degeneracy of energy level $E_n$. Sufficient time averaging is implied to get rid of the oscillations. We then define the degeneracy at $\beta$ as

$$d(\beta) := \frac{g(\beta, \infty)}{\mathcal{Z}(\beta)}. \tag{179}$$

The $\beta \to 0$ limit is the true average degeneracy across the entire spectrum while finite $\beta$ probes the degeneracy at finite energies.

# E  Haar Random Unitaries

Here, we review the operator entanglement in random matrix theory where the time-evolution operator is simply a Haar random unitary matrix[14]. For simplicity and relevance to the rest of the paper, we consider the second Rényi entropy. The normalized state is

$$|U(t)\rangle = \frac{1}{N^{1/2}} U_{i,\bar{j}}, \tag{180}$$

with $U$ being a Haar $N \times N$ dimensional matrix. The corresponding density matrix is

$$\rho = \frac{1}{N} U_{i,\bar{j}} U^*_{\bar{j}',i'}. \tag{181}$$

To normalize, we took the trace and average over the Haar group

$$\overline{\mathrm{Tr} U_{i,\bar{j}} U^*_{\bar{j}',i'}} = \int [dU] U_{i,\bar{j}} U^*_{\bar{j},i} = N. \tag{182}$$

This is a specific case of the general result for moments of the Haar measure [103]

$$\int [dU] U_{i_1,j_1} U_{i_2,j_2} \ldots U^*_{i'_1,j'_1} U^*_{i'_2,j'_2} \cdots = \sum_{\sigma,\tau \in \mathcal{S}_n} \delta_{i_1 i'_{\sigma(1)}} \ldots \delta_{i_n i'_{\sigma(n)}} \delta_{j_1 j'_{\tau(1)}} \ldots \delta_{j_n j'_{\tau(n)}} \mathrm{Wg}(N, \sigma\tau^{-1}), \tag{183}$$

where Wg is the Weingarten function. To compute the Rényi entropy, we partition the density matrix and take the partial trace

$$\rho_{AB} = \frac{1}{N} U_{i_A,i_{\bar{A}},\bar{j}_B,\bar{j}_{\bar{B}}} U^*_{\bar{j}'_B,\bar{j}_{\bar{B}},i'_A,i_{\bar{A}}}. \tag{184}$$

The average purity is then

$$\begin{aligned}
\overline{\mathrm{Tr} \rho^2_{AB}} &= \frac{1}{N^2} \int [dU] U_{i_A,i_{\bar{A}},\bar{j}_B,\bar{j}_{\bar{B}}} U^*_{\bar{j}'_B,\bar{j}_{\bar{B}},i'_A,i_{\bar{A}}} U_{\bar{j}'_B,\bar{j}_{\bar{B}},i'_A,i_{\bar{A}}} U^*_{i_A,i_{\bar{A}},\bar{j}_B,\bar{j}_{\bar{B}}} \\
&= \frac{-\frac{d_A d_{\bar{B}} + d_{\bar{A}} d_B}{\sqrt{d_A d_{\bar{A}} d_B d_{\bar{B}}}} + d_A d_B + d_{\bar{A}} d_{\bar{B}}}{d_A d_{\bar{A}} d_B d_{\bar{B}} - 1},
\end{aligned} \tag{185}$$

where the $d$'s are the Hilbert space dimensions of the sub-Hilbert spaces. This formula is exact and does not assume large $N$. The only two Weingarten functions needed were

$$\mathrm{Wg}(N, e) = \frac{1}{N^2 - 1}, \quad \mathrm{Wg}(N, SWAP) = -\frac{1}{N(N^2 - 1)}. \tag{186}$$

The average Rényi entropy is thus[15]

$$S^{(2)}_{A \cup B} = -\log \left[ \frac{-\frac{d_A d_{\bar{B}} + d_{\bar{A}} d_B}{\sqrt{d_A d_{\bar{A}} d_B d_{\bar{B}}}} + d_A d_B + d_{\bar{A}} d_{\bar{B}}}{d_A d_{\bar{A}} d_B d_{\bar{B}} - 1} \right]. \tag{187}$$

---

[14]Similar computations have appeared in Refs. [21, 33, 101, 102].

[15]The Haar average does not necessarily commute with the logarithm needed to define the Rényi entropy. For a more precise statement about the Rényi entropy, we would need a second replica trick taking case of the logarithm. However, this is not needed for sufficiently large Hilbert space dimensions where the operations approximately commute.

Taking linear combinations of the entropy, we find the operator mutual information

$$I^{(2)}(A, B) = \log\left(\frac{-d_A^2 d_B^2 + d_A^2 + d_B^2 - N^2}{d_A d_B - d_A d_B N^2}\right) + \log(d_A) + \log(d_B) \tag{188}$$

and tripartite mutual information

$$I_3^{(2)}(A; B_1, B_2) = -2\log(d_A) + \log\left(\frac{d_{B_2}^2\left(d_A^2 + d_{B_1}^2 - 1\right) - d_A^2}{d_{B_1}^2 d_{B_2}^2 - 1}\right)$$
$$+ \log\left(\frac{d_A^2\left(d_{B_1}^2 - 1\right) + d_{B_1}^2\left(d_{B_2}^2 - 1\right)}{d_{B_1}^2 d_{B_2}^2 - 1}\right). \tag{189}$$

Note that the first term is the infinite temperature result corresponding to maximal scrambling while the following terms are finite size corrections.

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
