# Peer review of "Information Scrambling with Conservation Laws"

_SciPost Physics, doi:SciPost Phys. 12, 117 (2022)_

## Round 1 · Referee Report · Anonymous (Referee 1) · 2021-9-12

Strengths

1- detailed discussion of scrambling and several new results in physically relevant situations. 2- investigation of the effect of conserved quantities on scrambling.

Report

The authors investigate the scrambling of information in several physically relevant situations, namely holographic conformal field theories and the Sachdev-Ye-Kitaev model. The authors focus on the situation in which the system possesses a conservation law. They argue that while energy conservation is compatible with maximal scrambling of the information, the presence of a U(1) conserved quantity lowers the amount of scrambled information. They also discuss the scrambling of operators.

Understanding the scrambling of information is an interesting and challenging topic that has attracted the attention of several communities. The authors address this problem by using several of the techniques that are currently available. Their results are scientifically sound and interesting. I think that this paper represents a useful addition to the present literature and I recommend its publication in Scipost Physics

I have some comments regarding the relation between this work and the literature:

1) Scrambling of information is known to appear also in integrable systems. This is because although in integrable systems the locality of information is preserved by the presence of well-defined quasiparticles, these have a nonlinear dispersion, unlike CFT systems. The authors could try to link their work to the results presented in

Phys. Rev. B 100, 115150 (2019)

2) The authors use the so-called equilibrium approximation to obtain the Renyi entropies of a state and of the thermofiele double. This amounts to saying that the Renyi entropies are the same as the Renyi entropies of the equilibrium state (formula (18) in the paper). This result holds generically also for integrable systems where there is an extensive number of conservation laws. The authors could mention

Phys. Rev. B 96, 115421 (2017)

---

## Round 1 · Referee Report · Anonymous (Referee 2) · 2021-9-25

Strengths

1- Timely topic. 2- New perspective.

Weaknesses

1- The equilibrium approximation seems too strong for locally interacting systems.

Report

The paper studies the tripartite operator mutual information (TOMI), a well-established measure of scrambling of quantum information, in systems with degeneracies in the spectrum. The starting point is an observation made in Ref. [15]: surprisingly, holographic CFTs do not saturate the well-known lower bound for the TOMI, indicating that they do not scramble quantum information maximally. In the paper the authors use the so called "equilibrium approximation" to argue that this is due to the large degeneracies in the spectrum of CFTs (also holographic ones) imposed by the structure of the Virasoro algebra. They propose degeneracies in the spectrum as a general mechanism to limit the scrambling of quantum information. To substantiate this picture they show that in the SYK model (and its charged variant), where the degeneracy is only twofold, the bound is saturated in at large $N$ (number of particles). Finally, they use their results to give a different perspective on the operator growth.

I think that the paper is interesting, it presents an overall clear and comprehensive discussion, and represents a non-trivial addition to the existing literature. Therefore, I am inclined to recommend publication in Scipost. Before that, however, the authors should clarify better the main approximation underlying their work, i.e. the equilibrium approximation. With that approximation they require the field configurations on one spacetime sheet to exactly coincide with that on another (backward evolving) spacetime sheet for all $x$ and $t$. Isn't this too strong? I would expect that also configurations where this happens only locally in the space-time would contribute non-trivially. Namely I would expect something like $\phi_i(x,t)=\phi_{\sigma_{x,t}(i)}(x,t)$ where also the permutation depends on the (coarse grained) space-time point. This is, for example, what happens in the discrete analogue of the problem discussed in Ref. [31]. Of course this complicates a lot the analysis: The problem becomes that of calculating the partition function of a Stat. Mech. model in 2d, rather than than in 0d. I can understand that this approximation can work for the SYK (as the authors show), but I struggle to believe that it is good enough for locally interacting systems.

Requested changes

The authors should address the point above and the following list of minor points.

1- In the introduction the authors refer to quantum circuit models as ``stochastic". Even if it's true that most studies have been conducted in random quantum circuits, which are indeed stochastic, this is not necessarily the case (for example the "dual-unitary" circuits studied e.g. in Ref.~[28] are not necessarily stochastic). I would suggest to use the term: periodically driven. Note that this is enough not to have energy conservation, which is the property the authors are interested in.

2- Why the regions $C$ and $D$ have to be semi-infinite in the discussion after Eq. 31?

3- I don't understand the claim "The only dynamical input thus far is that we are working with theories that have sufficiently complex energy spectra in order to decohere at late times. This situation is generic and should only be violated by integrable systems." made at the beginning of Sec. 3. Integrable systems (even free systems!) generically have a complex enough spectrum to decohere, see e.g.

Essler and Fagotti, J. Stat. Mech. (2016) 064002.

The only difference is that the diagonal ensemble (or thermomixed double) will have different properties, i.e. will be equivalent to a generalised Gibbs ensemble.

4- I am not sure the authors explicitly say that $\ell_A=\ell/2$ in Eq. 67 and in the inline equation before it.

5- Few typos:
"of" missing in the last sentence of the abstract.

"." should be "," in Eq. 41

"subscripts" $\mapsto$ "superscripts" in the sentence before Eq. 134.

---

## Round 2 · Referee Report · Anonymous (Referee 2) · 2022-2-18

Report

The authors satisfactorily addressed all my comments. I think the paper is ready for publication.

---

## Round 2 · Referee Report · Anonymous (Referee 1) · 2022-2-21

Report

I have read the new version of the manuscript. I thank the authors for considering the points that I suggested in my previous report. I think that the paper can now be published.

---

## Round 2 · Author Response

We thank both referees for their comments on our manuscript and their recommendation for publication. Below we address their specific comments and summarize the revisions we have made. ("R:" indicates the referees' comments and "A:" our responses).

Referee 1:

R: The authors investigate the scrambling of information in several physically relevant situations, namely holographic conformal field theories and the Sachdev-Ye-Kitaev model. The authors focus on the situation in which the system possesses a conservation law. They argue that while energy conservation is compatible with maximal scrambling of the information, the presence of a U(1) conserved quantity lowers the amount of scrambled information. They also discuss the scrambling of operators.

Understanding the scrambling of information is an interesting and challenging topic that has attracted the attention of several communities. The authors address this problem by using several of the techniques that are currently available. Their results are scientifically sound and interesting. I think that this paper represents a useful addition to the present literature and I recommend its publication in Scipost Physics

A: We thank the referee for their recommendation.

R: I have some comments regarding the relation between this work and the literature:

1) Scrambling of information is known to appear also in integrable systems. This is because although in integrable systems the locality of information is preserved by the presence of well-defined quasiparticles, these have a nonlinear dispersion, unlike CFT systems. The authors could try to link their work to the results presented in

Phys. Rev. B 100, 115150 (2019)

A: It is worthwhile to contrast the notion of scrambling in the suggested paper and the notion used in ours. In the above paper, the nonlinear dispersion of quasiparticles leads to information initially localized to broaden. However, in these integrable systems, the quasiparticles are still either purely left-moving or right-moving such that the information in a non-compact system is never delocalized across the entire system. A consequence of this fact is that the tripartite operator mutual information that we study for systems with quasiparticles will always be zero. In contrast, the systems we study that scramble strongly do not support quasiparticles and have an extensive amount of information delocalized across the entire system. We have added this discussion in a footnote on page 4.

R: 2) The authors use the so-called equilibrium approximation to obtain the Renyi entropies of a state and of the thermofiele double. This amounts to saying that the Renyi entropies are the same as the Renyi entropies of the equilibrium state (formula (18) in the paper). This result holds generically also for integrable systems where there is an extensive number of conservation laws. The authors could mention

Phys. Rev. B 96, 115421 (2017)

A: We stress that equation (18) in the paper is a formula for the \textit{fine-grained} entropy of the density matrix representing a subsystem of a pure state and is not a thermodynamic entropy i.e. subsystem of a thermal state akin to the ones studied in the suggested reference. This formula is a consequence of the equilibrium approximation and we are not aware of/do not expect this formula to hold for integrable systems.

We have added a footnote on page 12 mentioning that integrable systems do also decohere (but not to the extent that chaotic systems do) and included the suggested reference here.

Referee 2:

R: The paper studies the tripartite operator mutual information (TOMI), a well-established measure of scrambling of quantum information, in systems with degeneracies in the spectrum. The starting point is an observation made in Ref. [15]: surprisingly, holographic CFTs do not saturate the well-known lower bound for the TOMI, indicating that they do not scramble quantum information maximally. In the paper the authors use the so called "equilibrium approximation" to argue that this is due to the large degeneracies in the spectrum of CFTs (also holographic ones) imposed by the structure of the Virasoro algebra. They propose degeneracies in the spectrum as a general mechanism to limit the scrambling of quantum information. To substantiate this picture they show that in the SYK model (and its charged variant), where the degeneracy is only twofold, the bound is saturated in at large $N$ (number of particles). Finally, they use their results to give a different perspective on the operator growth.

I think that the paper is interesting, it presents an overall clear and comprehensive discussion, and represents a non-trivial addition to the existing literature. Therefore, I am inclined to recommend publication in Scipost. Before that, however, the authors should clarify better the main approximation underlying their work, i.e. the equilibrium approximation. With that approximation they require the field configurations on one spacetime sheet to exactly coincide with that on another (backward evolving) spacetime sheet for all $x$ and $t$. Isn't this too strong? I would expect that also configurations where this happens only locally in the space-time would contribute non-trivially. Namely I would expect something like where also the permutation depends on the (coarse grained) space-time point. This is, for example, what happens in the discrete analogue of the problem discussed in Ref. [31]. Of course this complicates a lot the analysis: The problem becomes that of calculating the partition function of a Stat. Mech. model in 2d, rather than than in 0d. I can understand that this approximation can work for the SYK (as the authors show), but I struggle to believe that it is good enough for locally interacting systems.

A: We first thank the referee for their positive evaluation of our work. The equilibrium approximation is, in fact, not at odds with the referee's intuition. At short times, before the system has globally equilibrated, field configurations in which the permutation depends on the space-time point, $\phi_{i}(x,t) = \phi_{\sigma_{x,t}(i)}(x,t)$, will indeed yield significant contributions to the path integral. As discussed in Ref. [31] of our original manuscript, domain walls will separate regions in space-time characterized by a specific permutation. These domain walls correspond to the ``entanglement membrane" discussed in Ref. [31] and Appendix A of our manuscript.

At short times, the dominant configuration of permutations and domain walls is given by the left panel of Fig. 6, in which membranes/domain walls stretch from the boundary of the input subregion to that of the output subregion. In the language used in the main text, the space-time region bounded by the membrane and region A is characterized by the identity $\eta$ permutation and the complement by the $e$ permutation (or vice versa, if the subsystem size is larger than half the system)

At late times when the system has thermalized, corresponding to the right panel of Fig. 6, the dominant configuration to the path integral is one in which the domain walls straddle the input and output subregions (yielding the volume law for entanglement). The bulk of the space-time is then dominated by the $e$ permutation. There will of course be other contributions to the path integral with additional domain wall configurations, but they will all be exponentially suppressed by the finite tension of the entanglement membrane. It is this late-time regime the equilibrium approximation addresses.

This is an important conceptual point and we have added a footnote on page 7 of our revised manuscript summarizing the above.

R: 1- In the introduction the authors refer to quantum circuit models as stochastic". Even if it's true that most studies have been conducted in random quantum circuits, which are indeed stochastic, this is not necessarily the case (for example the "dual-unitary" circuits studied e.g. in Ref. [28] are not necessarily stochastic). I would suggest to use the term: periodically driven. Note that this is enough not to have energy conservation, which is the property the authors are interested in.

A: We agree with the referee and have replaced stochastic" withperiodically driven" throughout.

R: 2- Why the regions $C$ and $D$ have to be semi-infinite in the discussion after Eq. 31?

A: This is not necessary and we have removed the word ``semi-infinite" in the discussion after Eq. 31. In the subsequent CFT computations, $C$ and $D$ end up being semi-infinite, but they are not semi-infinite in the SYK computations.

R: 3- I don't understand the claim "The only dynamical input thus far is that we are working with theories that have sufficiently complex energy spectra in order to decohere at late times. This situation is generic and should only be violated by integrable systems." made at the beginning of Sec. 3. Integrable systems (even free systems!) generically have a complex enough spectrum to decohere, see e.g.

Essler and Fagotti, J. Stat. Mech. (2016) 064002.

The only difference is that the diagonal ensemble (or thermomixed double) will have different properties, i.e. will be equivalent to a generalised Gibbs ensemble.

A: We were perhaps a bit loose in the use of the word ``decohere". As noted in the response to point 2 of Referee 1, we have modified the text and added a footnote to indicate that we restrict to systems which fully decohere i.e. equilibrate to the Gibbs ensemble rather than a generalized Gibbs ensemble. We have also included a reference to the mentioned paper here.

R: 4- I am not sure the authors explicitly say that $l_A = l/2$ in Eq. 67 and in the inline equation before it.

A: This was a typo. We have changed $l$ to $L_A$.

R: 5- Few typos

A: We thank the referee for pointing out these typos and have corrected them.

---

## Round 2 · List of Changes

1. Updated published references.

2. Replaced ``stochastic'' with ``periodically driven" in the abstract and introduction, as suggested in point 1 of Referee 2.

3. Removed the word ``semi-infinite" in the discussion after Eq. 31.

4. Corrected the typos noted by Referee 2 (points 4 and 5 in their report).

5. Added a footnote on page 7 to emphasize that the equilibrium approximation only holds at late times, to address Referee 2's remarks.

6. Modified a sentence on page 12 to no longer say that integrable systems do not decohere and added a footnote elaborating on this point. Here we also include the references suggested in point 2) of Referee 1 and 3) of Referee 2.

7. Added a footnote on page 4 to contrast different notions of scrambling.

---

## Editorial Decision

published